# Partitioning the primary ice formation modes in large eddy simulations of mixed–phase clouds

Luke B Hande[1] and Corinna Hoose[1]

[1]Karlsruhe Institute of Technology, Karlsruhe, Germany

*Correspondence to:* Luke B Hande (luke.hande@kit.edu)

**Abstract.**

State of the art aerosol dependent parameterisations describing each heterogeneous ice nucleation mode (contact, immersion, and deposition ice nucleation), as well as homogeneous nucleation, were incorporated into a large eddy simulation model. Several cases representing commonly occurring cloud types were simulated in an effort to understand which ice nucleation modes contribute the most to total concentrations of ice crystals. The cases include a completely idealised warm bubble, semi–idealised deep convection, an orographic cloud, and a stratiform case. Despite clear differences in thermodynamic conditions between the cases, the results are remarkably consistent between the different cloud types. In all the investigated cloud types and under normal aerosol conditions, immersion freezing dominates and contact freezing also contributes significantly. At colder temperatures, deposition nucleation plays only a small role, and homogeneous freezing is important. To some extent, the temporal evolution of the cloud determines the dominant freezing mechanism, and hence the subsequent microphysical processes. Precipitation is not correlated with any one ice nucleation mode, instead occurs simultaneously when several nucleation modes are active. Furthermore, large variations in the aerosol concentration do affect the dominant ice nucleation mode, however have only a minor influence on the precipitation amount.

## 1 Introduction

Ice crystals in the atmosphere can form spontaneously through homogeneous nucleation, which becomes increasingly probable at temperatures lower than -35°C (Koop and Murray, 2016). At warmer temperatures an ice nucleating particle (INP) is required to initiate freezing. Although INPs represent a small fraction of all atmospheric aerosols (Rogers et al., 1998), they have a disproportionately large influence on mixed–phase cloud microphysics (DeMott et al., 2010). Therefore modelling ice microphysical processes accurately is necessary to correctly model clouds, and the myriad subsequent processes influenced by clouds.

Several pathways have been identified through which ice nucleation in the atmosphere can take place (Vali et al., 2015). Deposition nucleation occurs at cold temperatures, where water vapour is deposited as ice directly onto an aerosol particle. Immersion and condensation freezing require the particle to be immersed in super–cooled liquid water, after which freezing occurs. Contact freezing occurs when an aerosol particle comes into contact with a super–cooled droplet, which subsequently initiates freezing. A similar mechanism called inside–out freezing has been identified, where a immersed particle comes into

contact with the water–air interface which initiates freezing (Durant and Shaw, 2005). Contact freezing and inside–out freezing have long been hypothesised to be important in areas of evaporation (Wang et al., 1978). Indeed, recent results from a modelling study support this idea (Hande et al., 2017).

Kanji et al. (2017) present a detailed overview of the latest ice nucleation research. Ice nucleation can be studied in a wide variety of ways (Cziczo et al., 2016), including under tightly controlled conditions in the laboratory. Recent reviews of laboratory experiments (Hoose and Möhler, 2012; Murray et al., 2012) highlight the tendency for much attention to be directed towards identifying and quantifying the ice nucleating ability of different aerosols species in each nucleation mode separately. These laboratory studies do little to elucidate the relative importance of these modes, so their atmospheric relevance is poorly understood.

Ladino et al. (2013) provide a review of experimental studies investigating contact nucleation, and go so far as to suggests it could dominate over immersion freezing for some aerosol species. Given that laboratory results suggest it is an efficient ice formation mechanism, these authors specifically pose the questions as to whether this also holds true in simulations. However in more recent experiments, Nagare et al. (2016) could not confirm a general enhancement in contact freezing compared to immersion freezing.

Modelling results from Cui et al. (2006) show that immersion freezing is the dominant pathway through which ice is formed, with contact playing little to no role. In this study, deposition nucleation was significant in the early stages of cloud develop-ment. Philips et al. (2007) used a model to also show that contact freezing has little impact on heterogeneous ice nucleation in deep convective clouds. An analysis of trajectories from a dust dominated region showed air parcels commonly pass through ice saturated, but water sub–saturated regions, where deposition nucleation could occur (Wiacek and Peter, 2009). Later, Hoose et al. (2010) showed that immersion freezing dominates INP production, and in contrast to the previous modelling studies, contact freezing played an important role in their simulations. Spichtinger and Cziczo (2010) used a model to show there is competition between heterogeneous and homogeneous ice nucleation, which is influenced by thermodynamic and microphys-ical conditions.

In–situ and remote sensing observations have also been employed to study ice nucleation under atmospheric conditions. Ansmann et al. (2009) observed altocumulus clouds which almost always had liquid water at cloud top, suggesting deposition nucleation plays little role. This has been supported by observations in cases of lee–wave clouds (Field et al., 2012) and stratiform clouds (De Boer et al., 2011; Westbrook and Illingworth, 2011), suggesting either immersion or contact freezing dominates ice production.

A recent global analysis of satellite observations (Carro-Calvo et al., 2016) indicates there are low cloud glaciation temper-atures in areas of deep convection, not only in the tropics, but also extending to the mid–latitudes. This suggests homogeneous freezing and/or deposition nucleation are important. The warm ice clouds analysed in their study, on the other hand, were associated with stratiform cloud systems, and the authors pose the question of the role that dynamics plays in initiating early cloud glaciation.

Since immersion and contact freezing require the presence of liquid water, they are thought to be the dominant ice formation pathway in mixed phase clouds. The above studies seem to suggest this is the case, however there is still considerable uncer-

tainty. In addition, there is little consensus on whether deposition nucleation or homogeneous freezing contribute significantly to ice production at cirrus temperatures.

A further complication arises since ice nucleation is clearly influenced by the ambient environmental conditions, and as such the dominant mode could depend on the cloud type. This paper aims to help clarify, in a systematic way, which ice nucleation modes dominate for various cloud types found over continental regions. The contribution of each mode to precipitation will also be considered. The cases studied here are a warm bubble, semi–idealised deep convection, idealised orographic, and a stratiform cloud, and hence cover a variety of thermodynamic conditions.

## 2 Model description

The non–hydrostatic regional weather forecasting model COSMO (COnsortium for Small-scale MOdelling) (Schättler et al., 2008) version 5.01 was run at high resolution of $8.9 \times 10^{-4}$ degrees ($\approx 100$ m). This scale is small enough to resolve energy-containing turbulence (Barthlott and Hoose, 2015). The two–moment cloud microphysics scheme of Seifert and Beheng (2006) was used, which uses the supersaturation to define a power law, from which CCN concentrations representative of continental conditions are calculated. The droplet size distribution was calculated from the model diagnosed cloud liquid water content and droplet number concentration in every grid box, assuming a modified gamma distribution, with parameters defined in Seifert and Beheng (2006), for droplets in the size range 1 to 535 $\mu$m. Figure 1 shows the spatial and temporal mean cloud droplet size distribution for each case investigated. These cases are described in detail in the next section.

Recent work has made significant progress in the development of detailed parameterisations for deposition nucleation, immersion freezing, and contact freezing (Niemand et al., 2012; Tobo et al., 2013; DeMott et al., 2010, 2014; Hiranuma et al., 2014; Steinke et al., 2014; Diehl and Mitra, 2015; Ullrich et al., 2017; Hande et al., 2017). These parameterisations were developed either from observations or theory, and are representative of nucleation on a variety of aerosol species.

In this study, the Steinke et al. (2014) parameterisation for deposition nucleation on Arizona test dust (ATD) was used. This parameterisation is a function of supersaturation with respect to ice, and temperature, and is active from 226–250 K. Niemand et al. (2012) was employed to describe immersion freezing, which depends on temperature, and acts between 237–261 K. In these two parameterisations, particle surface area also plays a role through the use of the ice nucleation active surface site (INAS) densities. Comparing these two parameterisations to recently developed formulations by Ullrich et al. (2017) shows good agreement for immersion freezing, and lower nucleation efficiency for desert dust compared to ATD. This provides some measure of confidence in the reliability of the parameterisations used here.

Hande et al. (2017) was used for contact freezing with generic dust aerosols. This parameterisation is a function of aerosol and droplet size and number concentration, relative humidity, temperature, and electrical charges, and is active between 240–268 K. Finally, theoretical expressions of the homogeneous nucleation rate by Jeffery and Austin (1997) were used to describe homogeneous freezing.

A two mode log–normal dust aerosol size distribution was used, as shown in Figure 1, covering particle sizes from 0.1 to 100 $\mu$m, which is based on observations from Jungfraujoch research station (Niemand, 2015) (Mode 1: N = 0.015$\times 10^6$ m$^{-3}$, $\mu$

= 1.355×10$^{-6}$ m, $\sigma$ = 1.443, mode 2: N = 0.00001×10$^6$ m$^{-3}$, $\mu$ = 8.518×10$^{-6}$ m, $\sigma$ = 1.358). Aerosol concentrations at sizes larger than about 30 $\mu$m are small enough as to be considered zero. The upper bound in the aerosol size distribution is only for mathematical convenience. The dust aerosol concentrations are constant in the vertical dimension throughout the simulation. Model results suggest that dust aerosols are relatively constant in the vertical dimension, with only a 25 % decrease of dust

aerosol number concentrations over Germany during summer between the low levels and the tropopause (Hande et al., 2015).

The aerosols are not removed by precipitation or sedimentation in the model. This simplification is not expected to have a significant effect on the formation of INPs. The maximum number concentration of aerosols is orders of magnitude larger than the maximum INP concentrations, as shown later in this manuscript. Therefore, any removal of aerosols will make a very small difference to the total number concentration. Furthermore, in the case of convectively or orographically forced clouds,

entrainment of new aerosols into the cloud adds a source of aerosols to off–set their removal. As for the stratiform case, a factor of 2 overestimate due to not depleting aerosols was found in a previous modeling study (Paukert and Hoose, 2014).

Hande et al. (2015) show that the 5$^{th}$ and 95$^{th}$ percentiles of dust number concentrations are representative of low and high dust concentrations. These concentrations are often more than an order of magnitude smaller and larger than the median, depending on the season. The dust aerosol properties used in this study correspond roughly to the properties during summer from

Hande et al. (2015), during which concentrations and aerosol sizes are the lowest throughout the year. In order to investigate the sensitivity of ice nucleation to the aerosol size distribution, two additional aerosol size distributions are defined in Figure 1, shown as the dashed lines. Here, the total number concentration of both modes was modified by a factor of $\pm 10$, which simulates high and low dust aerosol number concentrations.

The aerosol and droplet distributions were divided into 10 bins, over which the integration for the parameterisations was

performed. The immersion and contact freezing parameterisations are only applied to cloud droplets. Since rain drops collect many particles through collision–coalescence they may be important for freezing in the immersion mode, depending on cloud type (Paukert et al., 2017). However simple parameterizations for this process do not exist, limiting applicability of rain freezing through the immersion mode. Furthermore, Niehaus and Cantrell (2015) show that these deliquesced aerosol particles can initiate additional contact freezing.

Immersion freezing acts only on the immersed dust aerosols, and contact freezing acts on the interstitial aerosols. The segregation of immersed and interstitial aerosols is treated simplistically in this work, where the ratio of these quantities is pre–defined. In these simulations, 50% of the total number of dust aerosol is defined to be interstitial and hence available for contact freezing, and the remaining 50% is defined to be immersed and available for immersion freezing. This is not necessarily a realistic assumption, however it allows the relative concentrations of immersion and contact INPs to be compared independent

of this assumption, since differences in INP concentrations will not be due to differences in aerosol concentrations available for nucleation in a given mode. Finally, depletion of immersed aerosols is not taken into account in these simulations, which has been shown to cause an overestimate of the ice crystal concentrations by a factor of 2 for an arctic stratocumulus cloud (Paukert and Hoose, 2014).

## 2.1 Case study description

Ice nucleation is influenced by ambient environmental conditions, therefore in order to systematically study the relative contribution of each mode, a distinction between cloud types must be made. In this section, the model configurations for two cases of convection, an idealised orographic cloud, and finally a stratiform cloud, are described.

Since deep convective clouds span temperature ranges relevant for warm and cold cloud microphysics, including into the homogeneous nucleation regime, two cases will be investigated here: an fully idealised warm bubble case, and a semi–idealised cloud. Starting with the former, the thermodynamic profile described in Weisman and Klemp (1982) was used to initialise the simulation, shown in the left panel of Figure 2 as the black lines. A 3D temperature disturbance of 1.5 K, with radius of 10 km, was placed in the centre of the domain at a height of 1.4 km. 100 vertical levels, with $600 \times 600$ grid cells horizontally, were used, and the time step was 1 second for the duration of the 4 hour simulation.

The semi–idealised deep convective cloud represents a more realistic simulation of convection, and provides an interesting comparison with the previous idealised heat bubble. A detailed description of the model configuration for this case appears in Hande et al. (2017), and is summarised here. A real sounding with a convective available potential energy (CAPE) of 2801 J kg$^{-1}$ was used to initialise the simulation, and realistic topography was specified at each grid point, as shown in Figure 6 of Hande et al. (2017). The topography represents the region near Jülich, in western Germany, with mountains reaching up to 560 m in the south west of the domain. 100 vertical levels were used, and $600 \times 600$ grid cells horizontally, with a time step of 2 seconds for the duration of the 9 hour simulation.

To initialise the orographic mixed–phase cloud case, an idealised bell–shaped hill was used along with a real sounding, shown in the right panel of Figure 2 as the black lines. The hill has a maximum height of 800 m and a half–width of 15 km. In the longitudinal direction, 1441 grid points were used, and 271 in the latitudinal direction, with 100 vertical levels. A time step of 1 second was used for the duration of the 4 hour simulation.

The final case to be investigated is a stratiform cloud, which was initialised from a real sounding from central Germany during winter, shown in the right panel of Figure 2 as the blue lines. A smaller domain with $400 \times 400$ horizontal grid points were used, again with 100 vertical levels. In this case, the horizontal wind speed was artificially increased by a factor of 1.5 in the lowest 5.5 km, in oder to increase the dynamical forcing enough to activate cloud droplets through shear driven turbulence in the boundary layer. Due to the higher wind speeds in this simulation, a shorter time step of 0.5 seconds was used for the 9 hour simulation. All investigated cases employed fully periodic boundary conditions.

## 3 Spatial distribution of INPs

In this section the spatial distribution of INPs in each mode will be analysed, along with the cloud droplet properties. Contact freezing INPs are parameterised in terms of a rate, so multiplying by the time step of the simulation the number concentrations are obtained. All diagrams in this section are domain mean horizontal cross sections taken at a particular time step indicated in the figure captions, where the mean is taken over all latitudes. As described in the Section 2, cloud droplet size was calculated from cloud liquid water content and number concentration, assuming a gamma distribution at each grid point. The mode in the

cloud droplet radius distribution which is shown in the following diagrams is simply the radius at which the maximum in the cloud droplet size distribution occurred, and the variance and skewness of the distributions are not represented.

Starting with the idealised heat bubble, Figure 3 shows the concentrations of INPs (left panels), along with the cloud droplet properties (right panels) at 0.5 hrs into the simulation. Immersion and contact freezing both contribute significantly at warmer temperatures, and homogeneous nucleation is a major contributor at colder temperatures. Deposition nucleation, however, is limited to low concentrations occurring over a narrow temperature range.

Looking closer at immersion freezing, there is a trend of higher INP concentrations at colder temperatures. This should be expected since, according to this parameterisation, there is an inverse exponential relationship between INAS density and temperature.

Contact freezing, on the other hand, shows the opposite trend. Although the contact freezing efficiency also increases exponentially with decreasing temperature, droplet properties have a larger influence on INP concentrations, as discussed in Hande et al. (2017). The highest concentrations in the contact mode occur at around 6 km, co–located with the maximum in cloud droplet size. At colder temperatures above this height, the size and number concentration of cloud droplets is lower, reducing the effectiveness of contact freezing since the contact freezing collection kernel strongly favours large aerosol–large droplet interactions.

The final panel in Figure 3 shows the in–cloud relative humidity. On both sides of the central updraft, indicated by the solid contours, there are regions of downdrafts, shown by the dashed contours. This results in lower relative humidity which acts to suppress the formation of INPs.

The results for the semi–idealised deep convective case, shown in Figure 4, are remarkably consistent with the previous case: immersion and contact freezing both dominate, and homogeneous nucleation contributes the most at cold temperatures. Furthermore, the trend in immersion and contact INPs is the same as the idealised heat bubble.

The added complexity in this case highlights an interesting feature of the contact parameterisation employed in this study. Looking at the relative humidity, between about 16–26 km in the horizontal direction, the relative humidity is less than approximately 80 %. Despite this, the concentration of contact INPs are as high as $10^5$ m$^{-3}$. That INPs can still form in this environment is a consequence of the phoretic forces (Wang et al., 1978) increasing the collision efficiency between aerosols and cloud droplets in lower humidity regions. The lifetime of droplets can be calculated using Equation (3.14) from Houze (2014), ignoring curvature effects and assuming pure spherical droplets. A 10 $\mu$m droplet exposed to relative humidity of 80 % at 260 K should completely evaporate in 5.7 seconds, decreasing to 2.8 seconds at relative humidity of 60 %. Furthermore, Hande et al. (2017) show that in a deep convective cloud, droplets warmer than about 260 K can have number concentrations up to $10^8$ m$^{-3}$. These two points indicate there should be high numbers of droplets available for collisions within a few seconds before evaporating. Finally, another interesting feature of the deep convective case is the high levels of variability in INP concentrations along isotherms. This variability is attributable to the large influence of relative humidity and droplet properties on the contact freezing rate.

The orographic cloud case is shown in Figure 5. Here, homogeneous freezing and deposition nucleation play no role in ice formation, since the cloud top does not reach sufficiently cold temperatures, and immersion INP concentrations are significantly

higher than contact. Immersion INP concentrations are more–or–less homogeneously distributed throughout the cloud, and the highest concentrations in the contact mode are co–located with high concentrations of large cloud droplets. In the lee of the hill there is a down draft, indicated by the dashed contours. As was seen in the first case, this reduces the relative humidity and suppresses ice formation.

Given the different dynamical environment of the stratiform case, the resulting INP concentrations, shown in Figure 6, are quite low and the cloud is only sparsely populated with INPs, particularly in the immersion mode. Although the relative humidity in the mid–troposphere is high (around 60–70 %) compared to the other profiles shown in Figure 2, homogeneous freezing and deposition nucleation do not contribute to ice formation. Immersion INP concentrations are several orders of magnitude larger than contact INP concentrations.

The sounding used to initialise this case, shown in Figure 2, has a strong decrease in moisture at 5.5 km (T = 248 K, p = 475 hPa), which inhibits INP formation at colder temperatures. The maximum in the cloud droplet number concentration and size is between 1–2 km, which is outside the temperature range of the contact parameterisation. Therefore, the concentration of contact INPs is reduced due to the lower concentration of smaller cloud droplets in the region of contact freezing.

## 4    Temporal evolution of INPs

The temporal development of the ice phase influences a host of cloud properties, including cloud lifetime, radiative properties, and precipitation amount. Figure 7 shows the evolution of each INP mode over the duration of the idealised heat bubble simulation, where the domain mean concentrations over all latitudes and longitudes are taken. INPs in the contact mode appear in low concentrations after 15 min. The cloud develops rapidly, producing high concentrations of INPs in the immersion and contact freezing modes, as well as through homogeneous freezing. Deposition nucleation also plays a role early in the

simulation. As the simulation progresses, the initial convective cell dissipates, and after about 2 hours the simulation enters somewhat of a steady state as secondary convection is initiated throughout the domain. Immersion freezing plays less of a role in later stages of the simulation, and all other modes persist with roughly constant concentrations.

   The bottom panel show the domain mean accumulated precipitation for the duration of the simulation. Precipitation is initiated after about 1 hour, and there is a break in precipitation coinciding with the dissipation of the main convective cell, with

steady precipitation resuming after 2 hours. Interestingly, both cases with higher and lower dust aerosol concentrations result in higher precipitation. By the end of the simulation, there is a maximum difference of about 20 % in the total precipitation. Correlation coefficients for the domain mean integrated INP concentrations in each mode, and the domain mean total precipitation were calculated, and the correlation coefficients were not significant to any sufficiently high level of confidence. The CCN are not influenced by the dust aerosol distribution used in the INP parameterisations.

As in the previous section, the results in the two convective cases are similar, with the temporal evolution of the INPs in the semi–idealised deep convective case closely mirroring the evolution in the idealised heat bubble case, as shown in Figure 8. In the semi–idealised convective case the evolution of the cloud is notably slower, reaching maximum INP concentrations after 4 hours, at which time immersion freezing dominates. Towards the end of the simulation contact freezing becomes more

significant. INPs produced at cold temperatures of less than about -35 °C reach their maximum late in the simulation, with the greatest contribution from homogeneous freezing.

Precipitation is initiated during the peak in ice formation, between 3.5 and 5.5 hours into the simulation. During this time period is when the immersion and contact INP concentrations reach their maximum, and when homogeneous and deposition nucleation begin to play a role. Perturbations to the dust aerosol concentrations give the opposite effect compared to the previous simulation. That is, both cases of lower and higher dust concentrations give slightly less domain mean accumulated precipitation throughout the simulation.

The temporal evolution of the orographic case, shown in Figure 9, indicates INP production begins in the contact mode soon after initialisation, followed 15 minutes later by the immersion mode. As the simulation progresses, the cloud gets a few hundred metres deeper, immersion INP concentrations get gradually higher and contact INP concentrations get gradually lower.

The total precipitation in the orographic case is much lower than the previous two cases. Here, precipitation begins after 0.5 hours and is light and steady for the duration of the simulation. In contrast to the previous cases, the changes in the aerosol concentrations give a systematic change in accumulated precipitation, where higher aerosol concentrations result in higher precipitation, and vice versa. The difference in accumulated precipitation at the end of the simulation is around $\pm 10$ %.

The initial development of the stratiform cloud is similar to that of the other cases, where contact INPs are produced first, followed by immersion mode INPs, as shown in Figure 10. The contact mode develops slowly over the whole simulation, and is limited to low concentrations. Immersion INPs are produced later, but with higher average concentrations, and the cloud is stable for the duration of the simulation.

For the stratiform case, the precipitation is the lowest amongst all the cases. The simulation with higher dust concentrations shows about 25 % more precipitation, despite minimal changes in droplet size and number concentration. The simulation with lower dust concentrations has a negligible impact.

## 5   Domain Mean INPs

The results thus far are strikingly consistent: immersion and contact freezing dominate at varying times in the simulations, and in the convective cases, homogeneous freezing dominates in the cirrus regime. To quantify this further, Table 1 shows the spatial and temporal mean INP concentrations in each mode, including homogeneous freezing, along with the relative contribution to the total INP concentrations. The aerosol sensitivity simulation for each case are also shown, with - (+) indicating lower (higher) dust aerosol concentrations. Furthermore, the contribution of each mode until the onset of precipitation ($> 0.05$ kg m$^{-2}$) is shown. The concentrations quoted here are domain wide averages, meaning non–cloudy grid points are included, in order to not bias the results towards short lived, high INP concentrations.

This confirms that immersion freezing is clearly the dominant INP production mechanism in all cases. Contact freezing plays a significant role in most simulations, accounting for up to 1/3 of the total INP concentration in the simulation with normal

aerosol concentrations. In the convective cases, homogeneous freezing contributes most at cirrus temperatures, and deposition nucleation plays little role.

Leading up to the onset of precipitation, contact plays a dominant role in the semi–idealised convective case and the orographic cloud case. This is since contact nucleation is often the first ice formation mechanism activated, and in these two simulations contributes significantly at early stages of cloud formation. In the other two cases, immersion freezing contributes only slightly more than the simulations with normal aerosol concentrations.

## 6  Discussion

The INAS density for immersion freezing depends inverse exponentially on temperature. At temperatures of around 248 K, in the middle of the temperature range for the Niemand et al. (2012) parameterisation, the INAS density approaches $10^{10}$ m$^{-2}$. This should give an activated fraction of around 0.1 (0.95) for dust aerosols with radius 1 (5) $\mu$m. Given that most dust aerosols are much larger than 1 $\mu$m, immersion freezing is efficient in these simulations.

According to Hande et al. (2017), the contact freezing parameterisation depends primarily on aerosol and droplet sizes. These authors show that the highest contact freezing rates are obtained when large aerosol particles ($\gtrsim 0.3$ $\mu$m) interact with large cloud droplets ($\gtrsim 30$ $\mu$m). Only at the very largest sizes is the frozen fraction one. In these simulations, droplets are mostly smaller than 20 $\mu$m, resulting in a contact nucleation rate orders of magnitudes smaller than the maximum possible.

Deposition nucleation as parameterised by Steinke et al. (2014) depends inverse exponentially on temperature and exponentially on ice supersaturation. However it is tightly constrained by observations, such that it is only active at ice supersaturated conditions within a 24 K temperature window. This strongly limits the number of deposition INPs produced in the simulations. The homogeneous freezing parameterisation, on the other hand, is not as tightly constrained, and therefore dominates INP production at cold temperatures.

Some studies do suggest that, in the presence of large aerosol concentrations, homogeneous freezing could be inhibited by heterogeneous INP formation (Philips et al., 2007). The results presented here show that in the cirrus regime deposition nucleation contributes very little to ice formation, despite the high number concentration of aerosols in this region. The difference between the concentration of homogeneously formed ice and deposition nucleation INP is several orders of magnitude. This indicates that deposition nucleation is not suppressing homogeneous freezing.

The effect of perturbations in the dust aerosol concentrations is complex, and depends on the cloud type. In the convective cases, increasing aerosol concentrations increases the relative contribution of immersion freezing by an almost equivalent amount. The other freezing modes then compensate, resulting in a decrease in their relative contribution. The opposite is also true. Decreasing concentrations of dust aerosol decreases the contribution of immersion freezing, while increasing the relative contribution of the other modes. Indeed, in the idealised heat bubble case, contact freezing becomes the dominant mode in low aerosol conditions. There are, however, two exceptions where complex non–monotonic responses are evident: in deposition nucleation in the deep convective simulation, and in contact freezing in the stratiform case.

A natural question arises as to the sensitivity to the thermodynamic profile used to initialise the simulations, and hence how generalisable the results are. Given that the two convective cases, which had vastly different thermodynamic profiles, produced very similar results, this suggests the relative contribution of the ice nucleation modes is more–or–less insensitive to the initial conditions in these cases. Notice that the droplet properties of both convective cases, shown in Figures 1, 3, and 4, are very similar. Fan et al. (2017), however, show that thermodynamics contributes significantly to cloud microphysical processes for orographic mixed–phase clouds. This suggests the sensitivity for non–convectively forced clouds could be larger.

The stratiform case study represents the only cloud type in this study which is weakly forced. Despite high levels of moisture above the main inversion, the conditions for homogeneous freezing or deposition nucleation were not met in this simulation. There is a fundamental difference between cirrus produced in different dynamical environments. In the convective cases, liquid water is lifted from the mixed–phase regime to colder temperatures, where it freezes. Since the stratiform case is weakly forced, the origin of the moisture is from higher altitudes. These two categories are known as either 'liquid origin cirrus' or 'in–situ cirrus' (Krämer et al., 2016; Luebke et al., 2016). Since the stratiform cloud investigated here has no cirrus, the dominant ice forming mechanism for this so–called 'in–situ cirrus' remains an open question.

A few of the assumptions built into the simulations may influence the results presented. The even separation of immersed and interstitial aerosols will most likely cause an overestimate of contact freezing, in particular in the updraft where the supersaturation is the highest, and immersion freezing could be more dominant. Unprocessed dust has low CCN activity (Kumar et al., 2011), whereas aged dust is more likely to be immersed. The effect of this uncertainty is, however, expected to be small compared to the orders of magnitude difference in INP number concentrations between the different nucleation modes. Also, neglecting contact freezing of rain droplets should not have a large influence on the dominant freezing mode in these simulations, however it could affect the precipitation formation (Paukert et al., 2017).

A final consideration concerning the aerosol species needs to be made. Aerosol composition has a large influence on nucleation ability in different temperature and supersaturation regimes. Hoose and Möhler (2012) show that biological aerosols have a high onset temperature in the immersion mode, and given that certain biological aerosols can have large INAS densities at these warm temperatures (Murray et al., 2012), this could represent an important contributor to ice nucleation. A similar distinction between different dust species could also be made, since soil dust, for example, is more ice active in the immersion mode (Steinke et al., 2016). Whether this has a significant impact on the dominant ice nucleation mode remains to be investigated.

## 7 Conclusions

A number of high resolution modelling case studies are presented in order to systematically investigate which ice nucleation modes dominate for a number of typical cloud types. The results indicate that immersion freezing dominates in all cases. Contact nucleation plays a significant role in most simulations, accounting for between about 2–33 % of total INP concentrations under the reference aerosol conditions. Deposition nucleation only contributes a fraction of a percent in the convective cases,

and homogeneous freezing accounts for up to 6 % of total ice crystal concentrations. However in the non–convective cases, no INPs were produced in the cirrus regime.

In the later stages of the convective clouds, homogeneous freezing became more important, and contact freezing dominated at warm temperatures. INP formation in the orographic and stratiform case reached a steady state soon after the formation of the cloud. The occurrence of precipitation is not correlated with any one ice nucleation mode, instead occurs at the same time as multiple ice nucleation modes, including homogeneous nucleation.

Since the results from the two convective cases were quite similar, this suggests ice nucleation could be insensitive to thermodynamical conditions in these cases. The main consequence of the much higher CAPE in the heat bubble case, compared to the semi–idealised deep convective case, was faster cloud development.

For the convective cases, perturbation in aerosol concentrations produced proportional changes in the relative contribution of immersion freezing INPs. The relative contribution of the other modes decreased. The opposite occurred for the orographic case, where the relative contribution of contact increased under higher aerosol concentrations, and immersion decreased. In the stratiform case, all aerosol perturbations produced relatively more immersion freezing INPs, and fewer contact INPs. This indicates aerosol conditions have a complex influence on the dominant ice nucleation mode.

The response of the precipitation to perturbations in aerosol concentrations is also complex, and each case exhibits a different response. For the heat bubble, increasing and decreasing aerosol concentrations leads to an increase in precipitation. The opposite is true for the semi–idealised deep convective cloud, where both aerosol perturbations result in a decrease in precipitation. The orographic case shows proportional changes in precipitation in response to changing the aerosol concentrations, and in the stratiform case the higher aerosol concentrations produce more precipitation, with lower concentrations having no impact. This indicates that, although aerosol concentration plays a role in modifying precipitation, it is not the sole contributor. There could also be complex feedbacks present, where changes dust aerosol concentrations change the amount of ice produced, which in turn changes the latent heat release. This would have an impact on both the amount of liquid condensate and also the dominant ice nucleation mechanism.

*Data availability.* The data is available from the corresponding author upon request.

*Competing interests.* No competing interests are present.

*Acknowledgements.* This work was funded by the Deutsche Forschungsgesellschaft through the research unit INUIT–2 (FOR 1525, HO4612/1–2). The authors would like to gratefully acknowledge Christian Barthlott (KIT) and Lulin Xue (NCAR) for assistance with the model configuration.

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



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

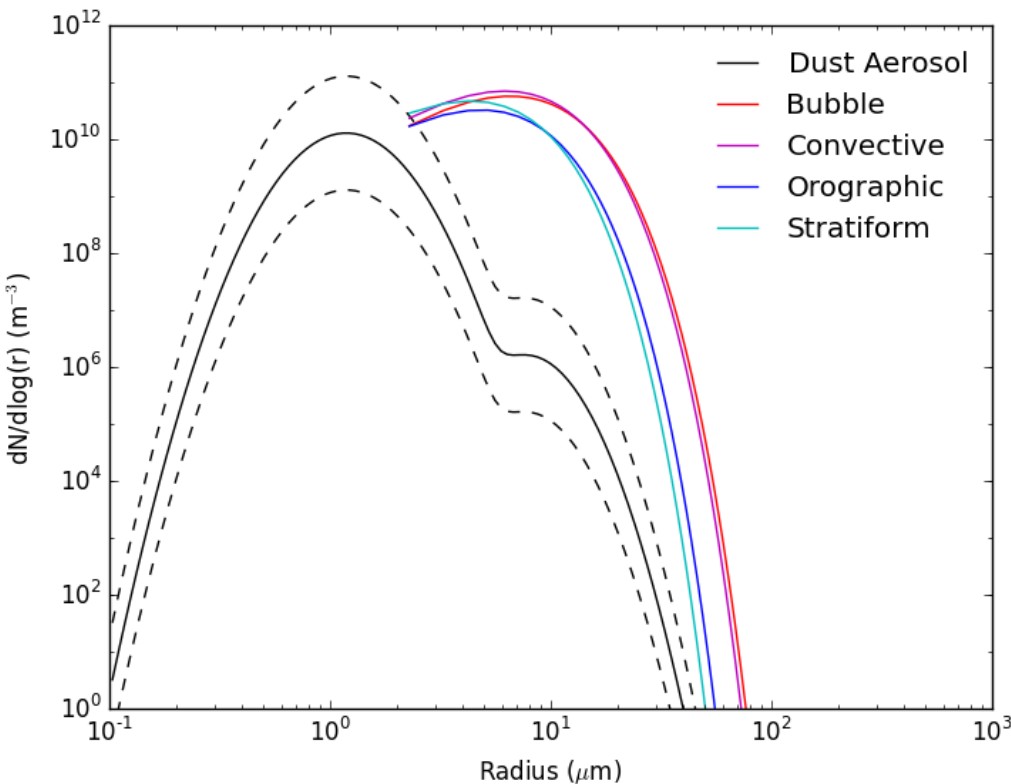

**Figure 1.** Prescribed dust aerosol size distribution, and derived mean cloud droplet size distribution for all cases. Dashed lines indicate dust aerosol size distribution for sensitivity studies.

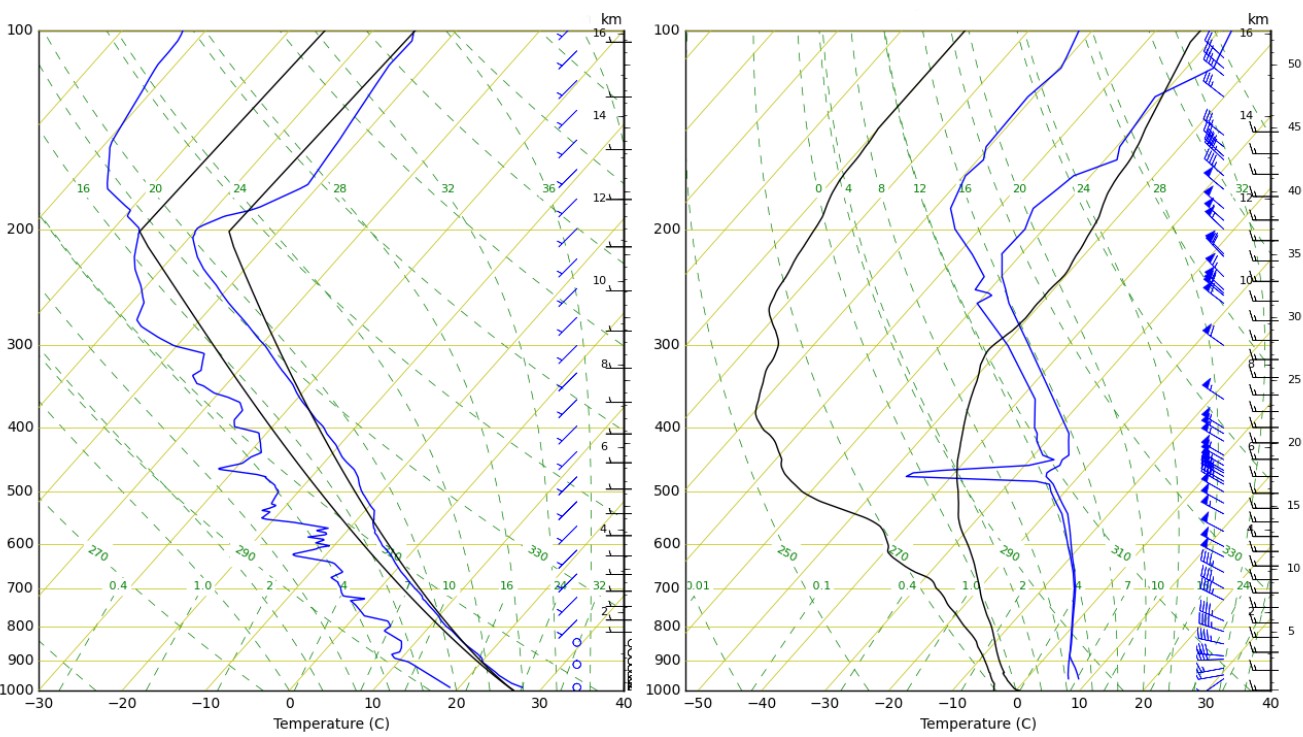

**Figure 2.** Thermodynamic sounding used to initialise the cases. Left panel: idealised heat bubble (black), semi–idealised deep convective (blue). Right panel: orographic (black), stratiform (blue).

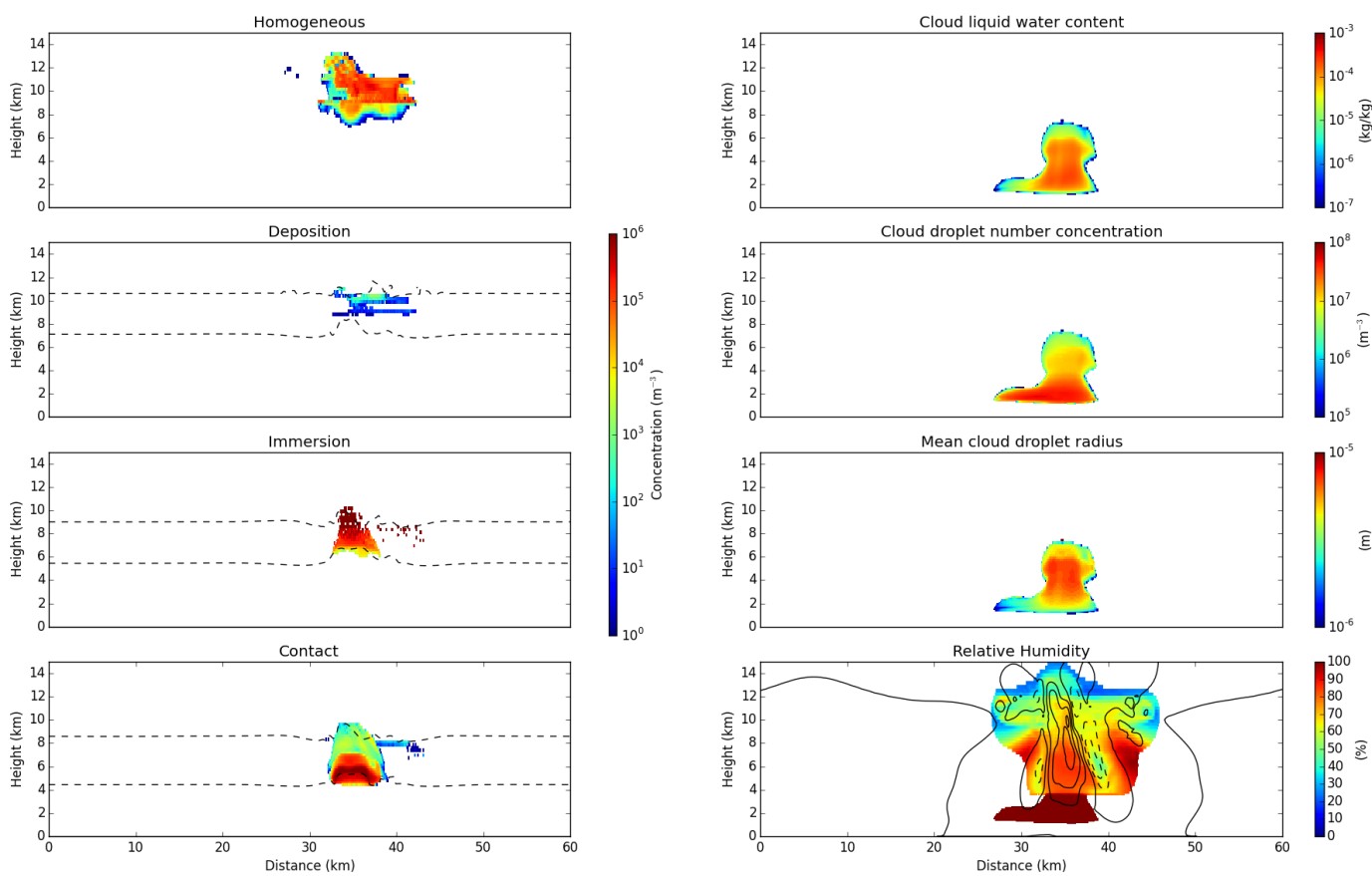

**Figure 3.** Domain mean horizontal cross section of INP number concentrations in each mode (left), cloud droplet properties (right) for the heat bubble convective cloud at 0.5 hrs into the simulation for normal dust concentrations. Dashed horizontal lines represent the temperature limits of the parameterisations. Contours represent the sign of the vertical velocity (solid: positive, dashed: negative).

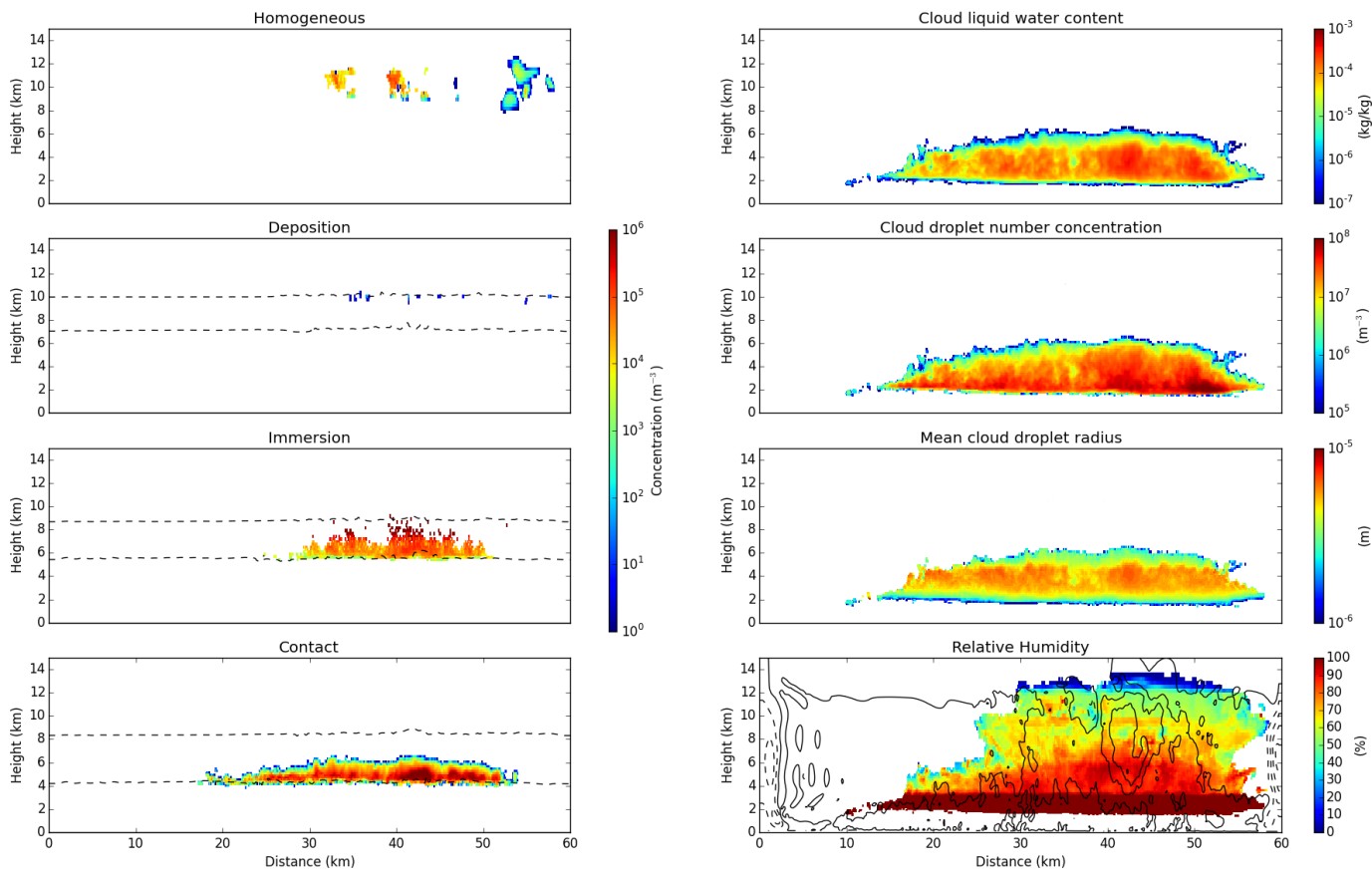

**Figure 4.** Domain mean horizontal cross section of INP number concentrations in each mode (left), cloud droplet properties (right) for the semi–idealised deep convective cloud at 4 hrs into the simulation for normal dust concentrations. Dashed horizontal lines represent the temperature limits of the parameterisations. Contours represent the sign of the vertical velocity (solid: positive, dashed: negative).

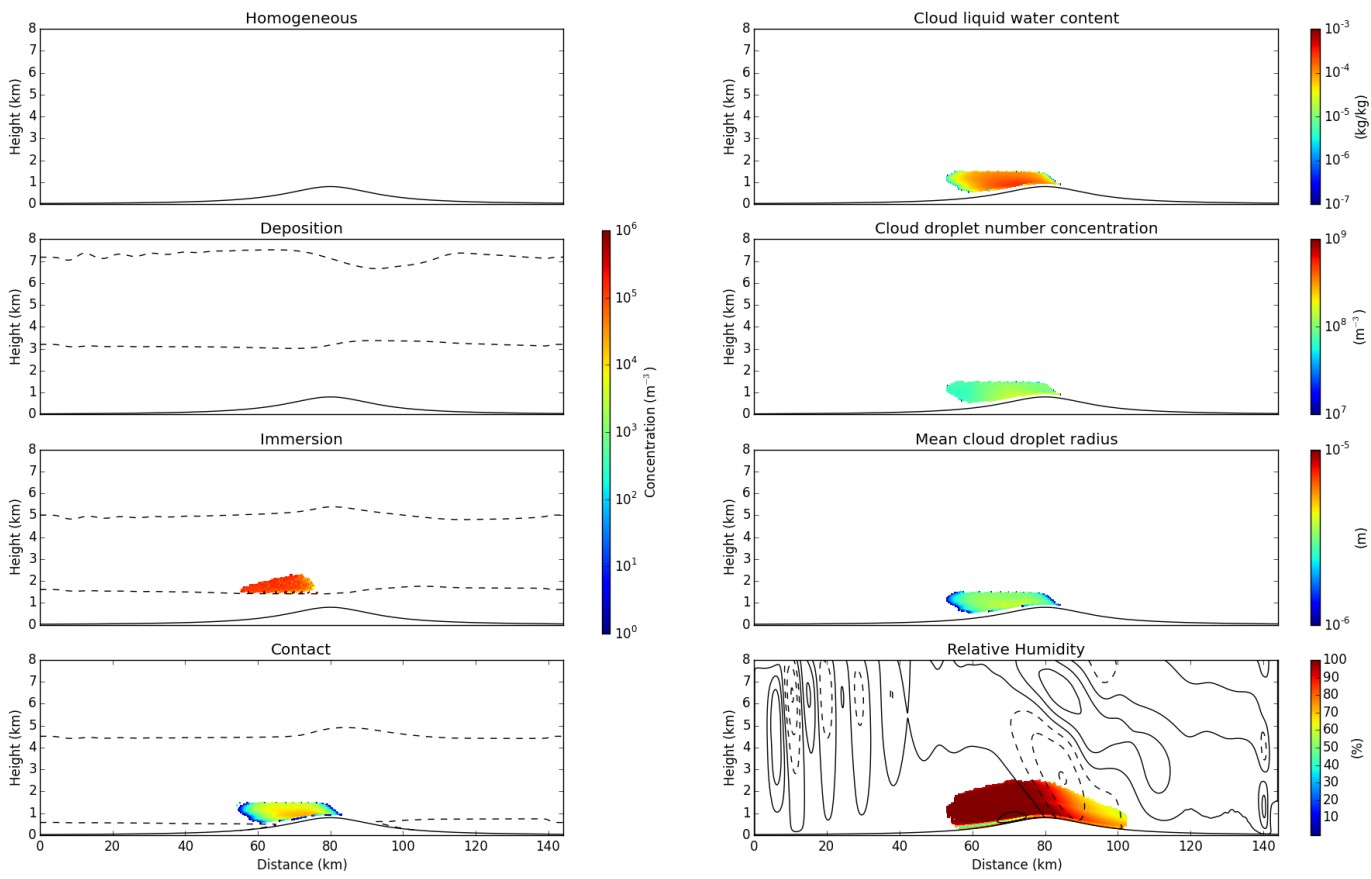

**Figure 5.** Domain mean horizontal cross section of INP number concentrations in each mode (left), cloud droplet properties (right) for the orographic cloud at 2 hrs into the simulation for normal dust concentrations. Dashed horizontal lines represent the temperature limits of the parameterisations. Contours represent the sign of the vertical velocity (solid: positive, dashed: negative).

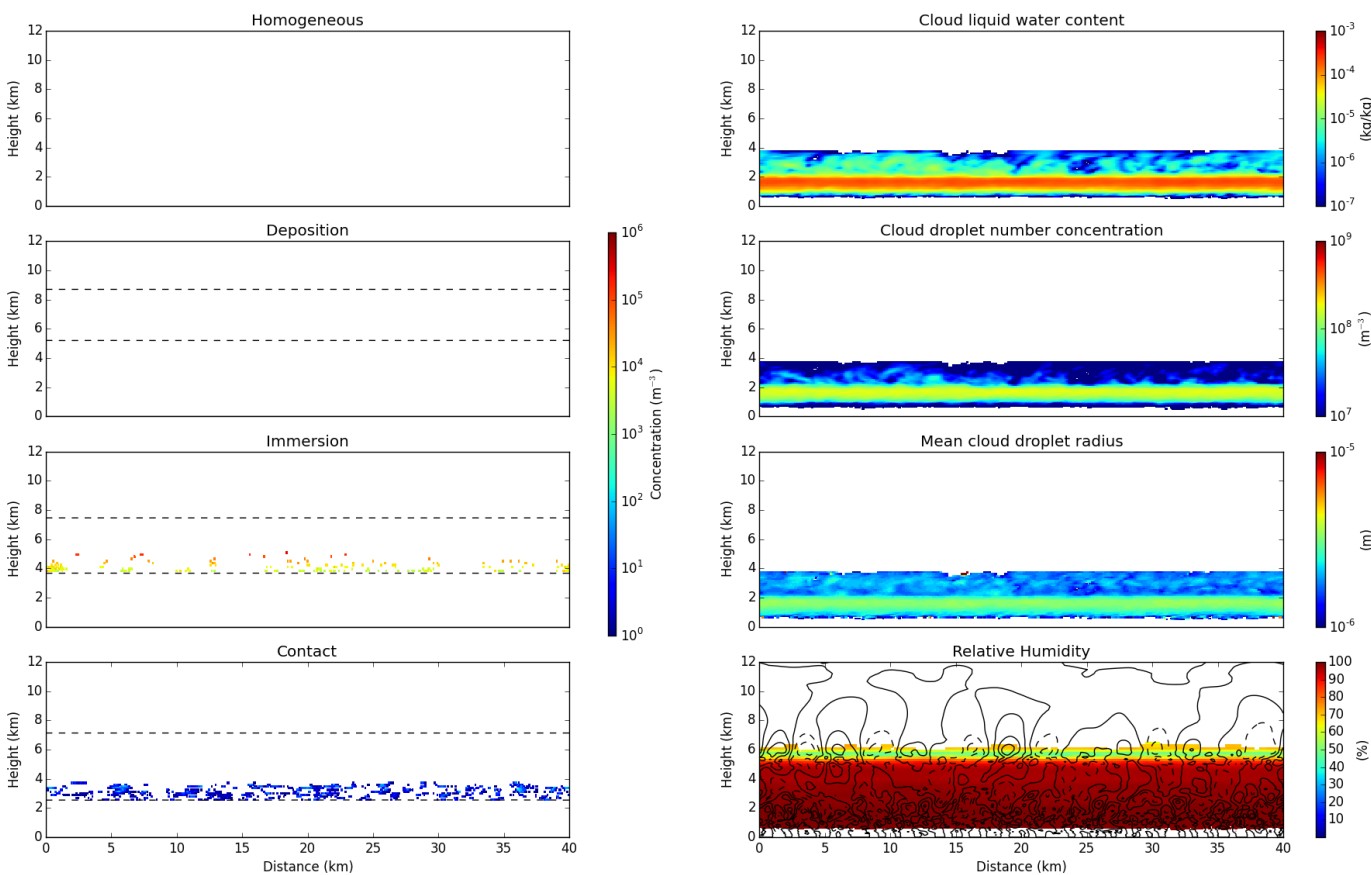

**Figure 6.** Domain mean horizontal cross section of INP number concentrations in each mode (left), cloud droplet properties (right) for the stratiform cloud at 3 hrs into the simulation for normal dust concentrations. Dashed horizontal lines represent the temperature limits of the parameterisations. Contours represent the sign of the vertical velocity (solid: positive, dashed: negative).

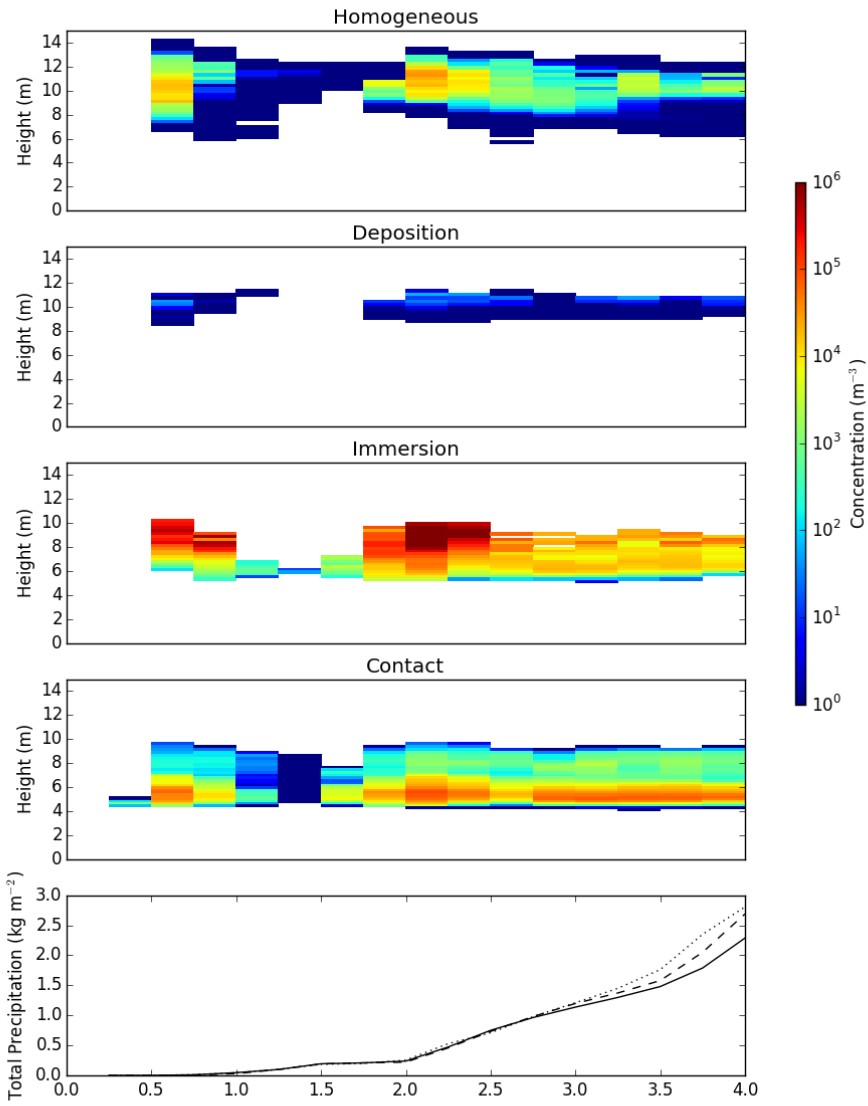

**Figure 7.** Temporal evolution of INP number concentrations in each mode for the heat bubble convective cloud for normal dust concentrations. Bottom panel shows total precipitation. Dashed (dotted) line is for the high (low) aerosol simulation.

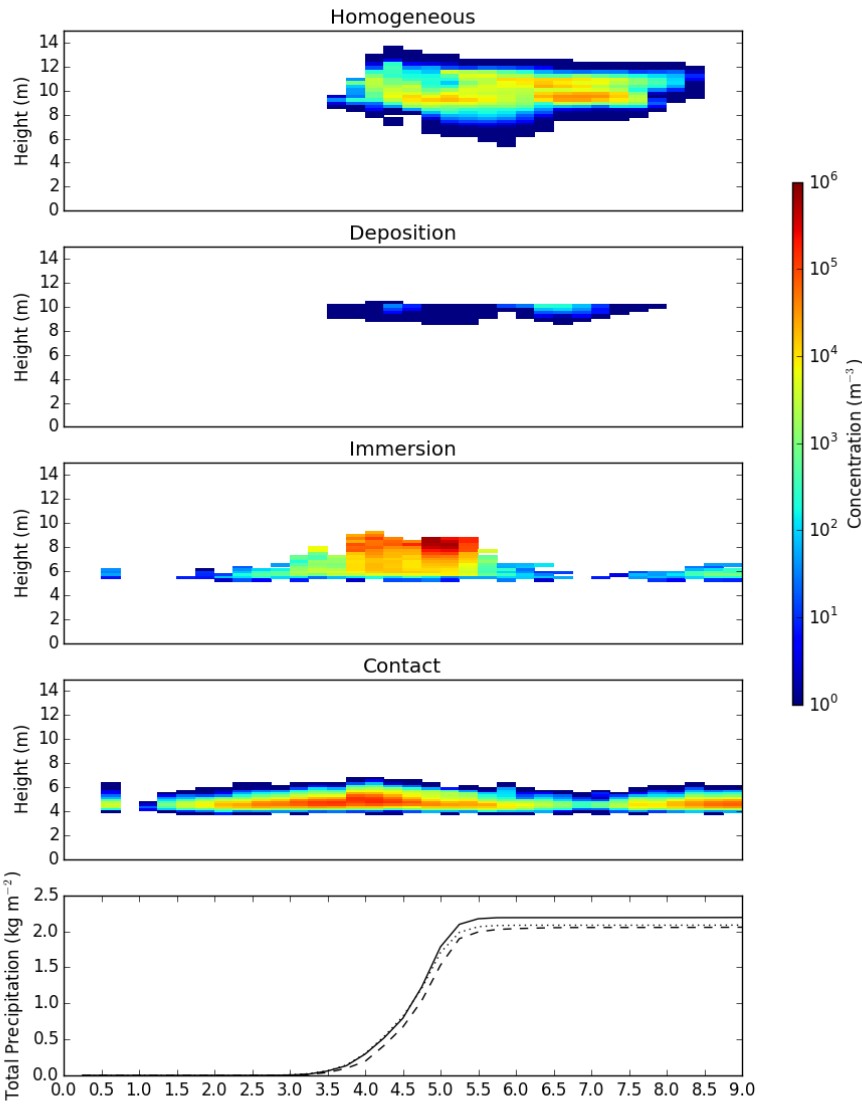

**Figure 8.** Temporal evolution of INP number concentrations in each mode for the semi–idealised deep convective cloud for normal dust concentrations. Bottom panel shows total precipitation. Dashed (dotted) line is for the high (low) aerosol simulation.

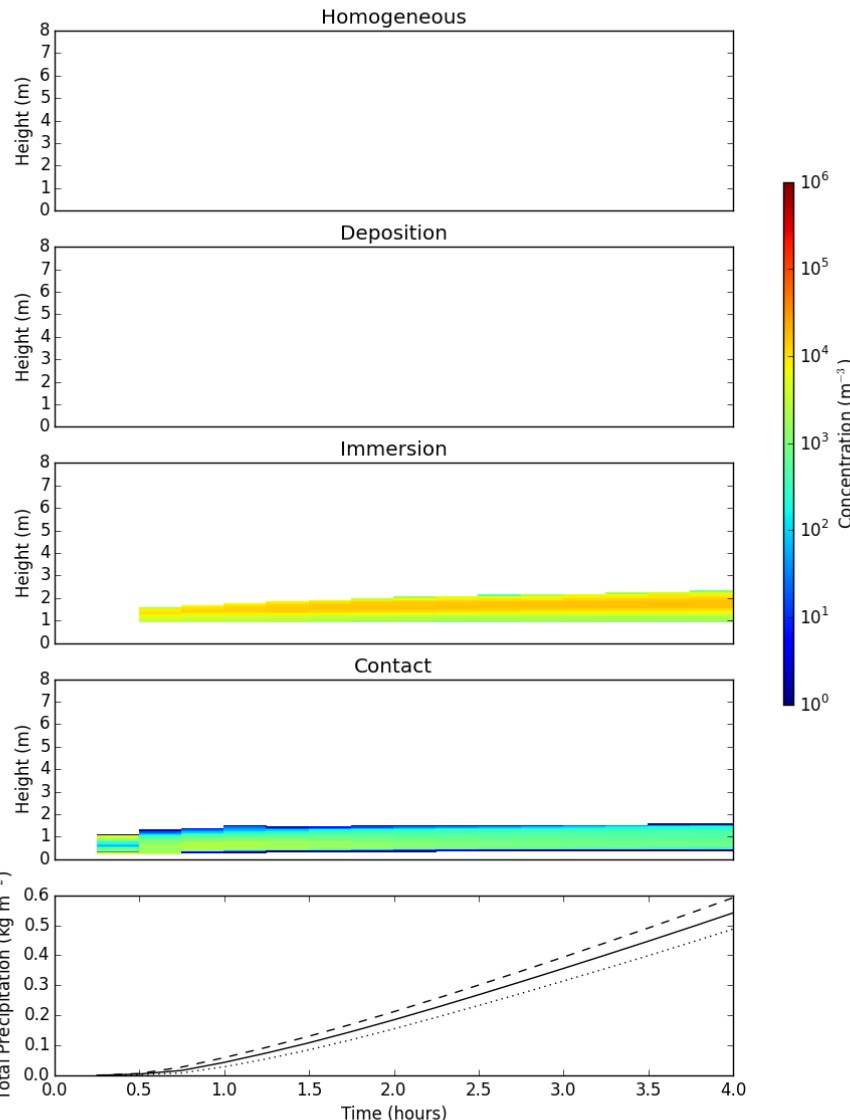

**Figure 9.** Temporal evolution of INP number concentrations in each mode for the orographic cloud for normal dust concentrations. Bottom panel shows total precipitation. Dashed (dotted) line is for the high (low) aerosol simulation.

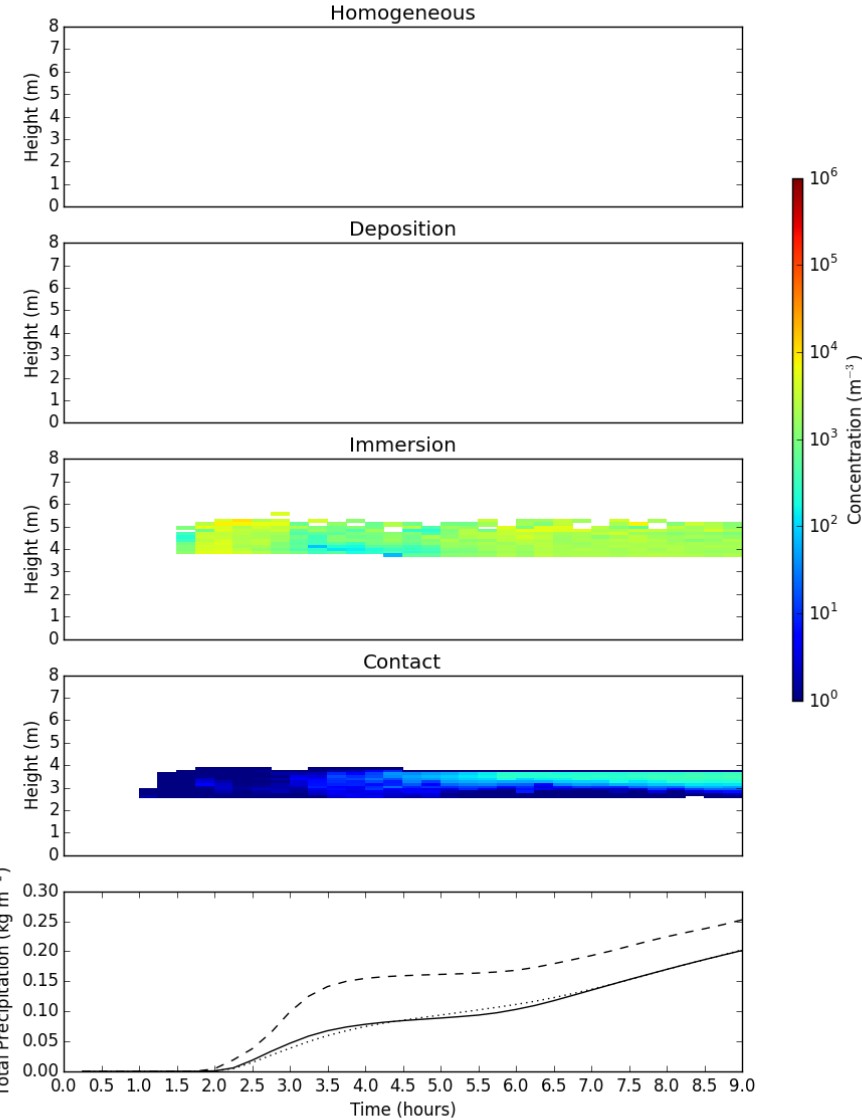

**Figure 10.** Temporal evolution of INP number concentrations in each mode for the stratiform cloud for normal dust concentrations. Bottom panel shows total precipitation. Dashed (dotted) line is for the high (low) aerosol simulation.

|  | Hom | Dep | Imm | Con |
|---|---|---|---|---|
| Heat Bubble + | $2.94 \times 10^2$ | $2.25 \times 10^0$ | $2.02 \times 10^5$ | $3.68 \times 10^3$ |
|  | (0.14) | (0.00) | (98.07) | (1.79) |
| Heat Bubble | $2.77 \times 10^2$ | $3.11 \times 10^{-1}$ | $1.76 \times 10^4$ | $2.05 \times 10^3$ |
|  | (1.38) | (0.01) | (88.31) | (10.30) |
| Heat Bubble - | $2.80 \times 10^2$ | $4.11 \times 10^{-2}$ | $2.09 \times 10^3$ | $8.43 \times 10^2$ |
|  | (8.73) | (0.00) | (65.07) | (26.20) |
| Heat Bubble Precip Onset | $3.41 \times 10^2$ | $2.14 \times 10^{-1}$ | $1.22 \times 10^4$ | $1.05 \times 10^3$ |
|  | (2.52) | (0.00) | (89.70) | (7.78) |
| Deep Convective + | $3.09 \times 10^1$ | $1.47 \times 10^{-1}$ | $2.37 \times 10^4$ | $2.43 \times 10^3$ |
|  | (0.12) | (0.00) | (90.56) | (9.32) |
| Deep Convective | $2.35 \times 10^2$ | $2.75 \times 10^{-1}$ | $2.11 \times 10^3$ | $1.24 \times 10^3$ |
|  | (6.56) | (0.01) | (58.95) | (34.48) |
| Deep Convective - | $2.51 \times 10^2$ | $8.73 \times 10^{-2}$ | $1.95 \times 10^2$ | $4.33 \times 10^2$ |
|  | (28.59) | (0.01) | (22.18) | (49.22) |
| Deep Convective Precip Onset | $3.14 \times 10^{-2}$ | $1.63 \times 10^{-5}$ | $3.97 \times 10^1$ | $9.88 \times 10^2$ |
|  | (0.00) | (0.00) | (3.87) | (96.13) |

| | | | | |
|---|---|---|---|---|
| Orographic + | 0 | 0 | $1.68\times10^3$ | $2.76\times10^2$ |
| | (0) | (0) | (85.84) | (14.16) |
| Orographic | 0 | 0 | $1.21\times10^3$ | $1.35\times10^2$ |
| | (0) | (0) | (89.91) | (10.09) |
| Orographic - | 0 | 0 | $7.65\times10^2$ | $4.32\times10^1$ |
| | (0) | (0) | (94.66) | (5.34) |
| Orographic Precip Onset | 0 | 0 | $5.77\times10^2$ | $2.08\times10^2$ |
| | (0) | (0) | (73.50) | (26.50) |
| Stratiform + | 0 | 0 | $6.21\times10^2$ | $1.81\times10^{-1}$ |
| | (0) | (0) | (99.97) | (0.03) |
| Stratiform | 0 | 0 | $1.87\times10^2$ | $4.73\times10^0$ |
| | (0) | (0) | (97.54) | (2.46) |
| Stratiform - | 0 | 0 | $1.85\times10^2$ | $1.49\times10^{-1}$ |
| | (0) | (0) | (99.92) | (0.08) |
| Stratiform Precip Onset | 0 | 0 | $1.76\times10^2$ | $8.37\times10^{-2}$ |
| | (0) | (0) | (99.95) | (0.05) |

Table 1: Temporal and spatial mean INP concentrations ($\mathrm{m}^{-3}$) for each case. + (-) indicates higher (lower) dust aerosol concentrations, as shown in Figure 1. The relative contribution (%) of each mode to the total INP concentrations is shown in parenthesis.