# Peer review of "Partitioning the primary ice formation modes in large eddy simulations of mixed-phase clouds"

_Atmospheric Chemistry and Physics, 2017_

## Referee Comment (RC1) · Anonymous Referee #1 · 1 Jul 2017

**Review of manuscript number acp-2017-499 by Hande and Hoose 2017**

**General Comments:**

Hande and Hoose present a study in which simulations of different cloud types in high resolution (LES simulations) with a variety of ice nucleation parameterizations representing different heterogeneous freezing modes and homogeneous freezing are used to elucidate the contribution of each freezing mode. This topic is of interest to the readers of ACP and the paper is generally well written. I would recommend the paper for publication in ACP after the following comments have been addressed.

I think the authors could do a better job of authors could do a better job of explaining the nuances of some observed aspects of their simulations. They are not over a page limit, and I think the paper can benefit from more explanations.

As it stands now, even though well written, it sounds more like a report with a few places in the manuscript where they apply an analysis of what the results mean. For example the authors make reference to steady state being achieved but without specifying with respect to which parameter?

Also, when there are effects of changing aerosol concentration, these are rightfully stated, but I think the authors could go one step further an explain why this would be expected to have influences (+ve or –ve biases) on the precip amount or total water content. On one hand, I understand that this is assumed knowledge, but on the other it would make the paper more round and complete.

Specific comments below:

**Page 1**

Line 1: maybe in parenthesis specify the modes contact, immersion and deposition so that it is clear that no other mode is being considered for instance evaporation freezing/PCF in presence of an active site etc.

Line 6: thermodynamical should read thermodynamic

Line 8: "little" should be replaced with "only a small"
Lines 11-12: here can you draw a connection between large aerosol variation and the mode of ice nucleation that would be dominant?

Line 15: I would say increasingly probable for temperatures lower than -35 C
Line 18: delete "in order"
Line 21: water vapour deposits directly to ice – you don't specify ice, you just mention deposition of water vapour which doesn't include a phase transition.
Line 25: "have long" should read "has long"

**Page 2**

Line 4: if talking about recent reviews then *Kanji et al.* [2017]is the most recent one that discusses the same topics mentioned in this paragraph.

Line 13-20 can be one paragraph. No need to have two paragraphs

Line 33 " delete "to this"
Line 34, doesn't the study by [*Spichtinger and Cziczo*, 2010] deserve mention here, since they looked into the competition between homogeneous freezing and heterogeneous ice nucleation (deposition nucleation)

**Page 3**
Line 5: thermodynamic conditions

Line 10: supersaturation is one word and everywhere else in the paper

Line 15-18 should be in the introduction – doesn't fit in the model description section

Line 30-31: dust size distribution at Jungfraujoch from 0.1 – 100 µm, I find that a little hard to believe. Do you have better reference for that? There are a lot of papers published on aerosol and INP properties at the Jungfraujoch that may give you a representative size distribution. I didn't think 100 um dust particles would make it from North Africa to central Switzerland, or at least not in any significant proportion.

Line 32-33: If dust is not removed by precipitation in the model, which should be one of the key removal processes – how does this affect ice nucleation in further time steps, i.e. dust that was at lower altitudes that did not activate (because T is not low enough) but also did not get removed by precipitation could then be available to be lofted or for convective uplift for next time steps? Wouldn't this positively bias the role of dust as INP in the model runs?

**Page 4**
Line 8-11. The authors state that contact freezing of rain drops is not considered but in the every same sentence say that rain drops collect many particles through collision-coalescence processes. But why should this be a reason for contact freezing not to be considered. Perhaps the collision processes could lead to freezing rather than coalescence? Also, in a recent study it was shown that a deliquesced surface of an aerosol particle colliding with a droplet can also induce contact freezing [*Niehaus and Cantrell*, 2015], therefore potentially enhancing the contribution of contact freezing

Line 14-15: this is 50% by number I assume? Please specify

Line 17: Depletion of immersed aerosols is not taken in to account in these simulations? Does this mean that aerosols that get immersed in cloud droplets are still available for CCN activation in further time steps? How does this impact the results obtained? Given that 50% of your aerosol (by number I assume) is immersed, how would this influence the outcome of the simulations if they are assumed to be available for CCN activation/immersion freezing in subsequent time steps?

Line 19: Replace "so" with "therefore" so is colloquial

Line 30: Has CAPE been defined before?

Line 9: x-hour simulation? Perhaps I misunderstood something?

**Page 5**

Line 28-30. I am confused by the wording and reasoning in this sentence. The smaller size and droplet number inhibit INP formation? Perhaps the authors meant "reduce the effectiveness of contact freezing" because of fewer collisions? More INPs would be expected to be active at colder temperatures, but it is fathomable that if collisions do not occur in the first place, then the role of contact freezing would be limited. Perhaps clarify, but the wording "formation of INP" sounds incorrect to me.

**Page 6**

Line 1-3: is there a chance to discuss here or comment on whether homogeneous freezing was suppressed or its initiation was suppressed because of the formation of ice heterogeneously and potentially depleting water vapour. This is hard to deduce from the way the figures are presented. One could discuss size of ice crystals here as well. Perhaps this is lower in the manuscript under temporal distribution..

Line 5-10: So you have droplets available for collisions with INPs at RH below 80% how long do they survive and how concentrated are they if they are surviving as droplets at such low RH?

Line 17-20: I don't get the reasoning here. The RH being high in the mid troposphere shuld warrant deposition and homogeneous freezing taking place? I would imagine high is a relative term here. 60-70% is high compared to what? So 60-70% RHw is low for homogeneous freezing to take place, so not the result being referred to here is not surprising. Based on lab studies, I am not surprised that deposition nucleation is not contributing at such low RHs either. Generally RHw 70% at about 223 K is usually required for deposition nucleation (or pore condensation and freezing).

**Page 7**

Line 16-17: it is not clear to me how the authors come to this conclusion about the precipitation. Is this in reference to the total precipitation over the course of the simulation for both cases being compared, or comparing precipitation at a given time stamp? Or the precipitation at the end of the simulation?

**Page 8**

Line 1-10: I like the table with the relative contributions of each freezing mode. I assume this is the total contribution over the course of the simulation. However, it would be nice to see more interesting versions, for example you could consider just quantifying the relative

contributions up to the point of precipitation initiation – if you just consider the simulation until precipitation starts, could you say something about how much each mode contributes to initiation of precipitation?

**Page 9**

Line 15: Reference format?

Line 15-18: I assume you are talking about different aerosol species. In this work you have tested dust parameterizations, but one could easily predict how this would change if an organic aerosol parameterization for ice nucleation or soil dust one was used? This should simply make the ice nucleation more or less effective (i.e. lower Ns in the case of organics compared to dust for a set T for example). Couldn't a quick statement from a simulation be made about that in this paper. At least the contribution from soil dust would have been interesting to see here.

Line 23: homogeneous freezing accounts for 6% of INP concentrations? Do you mean ice crystals, I didn't think homogeneous freezing was associated with INPs?

Line 26: steady state being referred to in terms of which quantities?

**Page 10**

Line 3: "on" the dominant ice nucleation mode?

Figures:

Are Figures 1-6 for high or low dust concentrations?

Figure 1: nm or $\mu$m is more intuitive. I do acknowledge you want to stick to SI units.

Figure 3. Homogeneous freezing sets in at about 8km but you have quite a number of liquid droplets above this altitude, are these meant to be conc. solution droplets because your RHw isn't that high? Can you please clarify the existence of liquid droples here? Referring to the top 2 panels in the right column.

Figure 4: Similar to Figure 3: should there be any droplets when homogeneous freezing has kicked in? Also for the mean cloud drop radius a small comment, more ticks on the scale could be helpful.

Figure 5: In all these panels, is it possible to zoom into the orographic cloud more and reduce white space .. i.e. there is nothing to show for the altitudes above 3 km - it would give a more clear picture of the orographic cloud. The parameterization limit lines are shown in the Figures 3 and 4, so the reader can be referred to the same limits in Figure 5.

Figure 6: Same comment as Figure 5.

Figure 10. Is this figure referred to in the text? I don't think so. Also, why the differences in total water content for the high dust case and lower precip for the lower dust case? perhaps explain this a bit more in the text.

**References:**

Kanji, Z. A., L. A. Ladino, H. Wex, Y. Boose, M. Burkert-Kohn, D. J. Cziczo, and M. Krämer (2017), Overview of Ice Nucleating Particles, *Meteorological Monographs*, *58*, 1.1-1.33, doi:10.1175/AMSMONOGRAPHS-D-16-0006.1.

Niehaus, J., and W. Cantrell (2015), Contact Freezing of Water by Salts, *J. Phys. Chem. Lett.*, *6*(17), 3490-3495, doi:10.1021/acs.jpclett.5b01531.

Spichtinger, P., and D. J. Cziczo (2010), Impact of heterogeneous ice nuclei on homogeneous freezing events in cirrus clouds, *J. Geophys. Res.-Atmos.*, *115*, D14208, doi:10.1029/2009jd012168.

---

## Referee Comment (RC2) · Anonymous Referee #2 · 15 Jul 2017

Review of "Partitioning the primary ice formation modes in large eddy simulations of mixed–phase clouds" by Hande and Hoose

The authors have conducted LES for a few typical cloud cases to systematically investigate which ice nucleation modes dominate. The study has some interesting findings, such as, immersion freezing dominates and contact freezing also contributes significantly in all cases. At colder temperatures, deposition nucleation plays little role, and homogeneous freezing is important; the temporal evolution of the cloud determines the dominant freezing mechanism; precipitation is not correlated with any one ice nucleation mode, instead occurs simultaneously when several nucleation modes are active. However, I have some major concerns which could impact the key conclusions of the paper. Addressing these concerns would make a more solid paper.

Major comments:
(1) The results could be dependent on how dust aerosol particles are treated in the model (i.e., prognostic or fixed during the simulation). The information about this is lacking. If dust aerosol particle size distribution is fixed as constant, then cloud properties would not be realistic, especially results of the temporal changes would not be reliable. If prognostic, the relative importance of different modes could depend on the calling order of these modes in the code. For example, if contact freezing is called before deposition freezing and it is efficient, then a lot of interstitial aerosols would be consumed at low-levels (warm temperatures), which would decrease the transported aerosols reaching the cold temperature so the less contribution of deposition freezing will be seen. However, if deposition is called before the contact freezing, the results could be totally changed. The authors did say the dust aerosol concentrations are constant in the vertical dimension. Is it just at the initial time or during the simulation. If it is during the simulation, how is it be realistic? Different assumptions of vertical distributions of dust aerosol particles could also impact the relative contributions of different freezing modes. Perhaps some sensitivity tests on this would like to gain some ideas.
(2) The key conclusions would also be affected by different assumption of the ratio between the immersed and interstitial aerosols. The even ratio used in this study needs some justification (any literature showing such a ratio would do). If there is no justification for this, the generalization of the key results of this paper can be questioned.
(3) More analysis is needed, either to support a key conclusion point or to explain the results (see specific comments # 1, 9, 14, 15). Very often, the authors only describe the results but not go further to explain and understand why.

Specific comments:

1) P. 1, Line 10-12, how about changes of rain rate PDF? Also, I did not see significant results presented for the point "Precipitation is not correlated with any one ice nucleation mode".

Since this is one of the key results, the corresponding correlation plots should be found easily in the result section.

2) P. 2, L14-15, which study? Also, suggest to use past tense consistently for describing what past work did. Currently, the author mixed the past tense with present tense for these descriptions.

3) P. 2, L22, I am confused here. How can deep convective clouds always have liquid water at cloud top? This might occur in mixed-phase clouds, but not the deep convective clouds.

4) P. 2, L32, need to clearly state which studies since no specific studies is mentioned yet in this paragraph.

5) Section 2, Model description: need more information about dust aerosol simulation in the model, for example, is the dust aerosol size distribution prognostic or fixed during the simulation? See my major comment #1 about the importance of the information. Also, about the vertical distribution, needs to clarify the constant concentration is just at the initial time or during the simulation.

6) P. 4, L15-17, I am not clear how the assumption of the ratio allows the relative concentrations of immersion and contact INPs to be compared independent of this assumption. In addition, is there any measurements in literature showing a ratio of immersed to interstitial aerosols in any place? This ratio could affect the relative contribution of different modes a lot. If there is no justification for this, the generalization of the key results of this paper can be questioned.

7) P.5, first paragraph, please describe that it is an orographic mixed-phase cloud case.

8) Figure 3, how to explain the two disconnected layers of liquid in this warm-bubble case? The layer between 9.5-13.5 km is not realistic. Temperatures in this layer could be lower than -38 $^0$C, so liquid particles generally can not survive. So, why is there no liquid between 7.5-9.5 km?

9) p.7, L17, why are both lower and higher aerosol concentrations giving less precipitation?

10) P. 7, last paragraph, the sentence "While the stratiform case has moderate amounts of liquid water, the total precipitation is the lowest amongst all the cases, so much so that the precipitating liquid doesn't decrease the total water" is confusing. The first part of the sentence does not mean much since all the cases are different type of clouds and comparing the correlation between liquid water and precipitation among different types of the clouds does not make much sense physically. Second, I am not sure what you really want to say here for the second part of the sentence "so much so that…".

11) P.8, Section 4, this is a discussion session. Table 1 clearly shows main results, so I would suggest to present it earlier (i.e., in the result section).

12) P9, L9, do you mean the stratiform cloud investigated here has no in-situ cirrus?

13) P9, L22, what is "a fraction of a percent"?

14) P9, L25-26, what makes "contact freezing dominated at warm temperatures"? Why immersion freezing is less contributed?

15) Last paragraph: Need to explain why the perturbation in aerosol concentrations (means increase or decrease of aerosols) produced proportional changes in the relative contribution of immersion freezing INPs and the relative contribution of the other modes decreased the convective cases. It is especially important to understand how all the other

modes are decreased for both increasing and decreasing aerosols. Also, what makes the different results of aerosol impacts among the convective, orographic, and stratiform cases?

Minor comments,

P. 1, L6, "in each case" should be "between the cases".

P9, L3, change "an orographic mixed–phase case" to "orographic mixed–phase clouds".

16) P9, L6, incomplete sentence: "There is a fundamental difference between cirrus produced in different dynamical environments", between cirrus and what?

---

## Author Comment (AC1) · 31 Aug 2017

Dear Editor,

The authors would like to thank you for handling the review of our manuscript. The authors would also like to thank the reviewers for providing constructive and timely reviews of the manuscript, which will result in a much improved publication. Below is a list of comments raised by all reviews, the authors response, and any manuscript changes as a result. Please note, that all line numbers refer to the original submitted version of the manuscript.

**REVIEWER #1:**

**General Comments:**
Hande and Hoose present a study in which simulations of different cloud types in high resolution (LES simulations) with a variety of ice nucleation parameterizations representing different heterogeneous freezing modes and homogeneous freezing are used to elucidate the contribution of each freezing mode. This topic is of interest to the readers of ACP and the paper is generally well written. I would recommend the paper for publication in ACP after the following comments have been addressed. I think the authors could do a better job of authors could do a better job of explaining the nuances of some observed aspects of their simulations. They are not over a page limit, and I think the paper can benefit from more explanations. As it stands now, even though well written, it sounds more like a report with a few places in the manuscript where they apply an analysis of what the results mean. For example the authors make reference to steady state being achieved but without specifying with respect to which parameter? Also, when there are effects of changing aerosol concentration, these are rightfully stated, but I think the authors could go one step further an explain why this would be expected to have influences (+ve or –ve biases) on the precip amount or total water content. On one hand, I understand that this is assumed knowledge, but on the other it would make the paper more round and complete.

Specific comments below:

**Page 1**
Reviewer Comment:
Line 1: maybe in parenthesis specify the modes contact, immersion and deposition so that it is clear that no other mode is being considered for instance evaporation freezing/PCF in presence of an active site etc.

Author Response:
Thanks.

Manuscript Changes:
Text added: '… (contact, immersion, and deposition ice nucleation) …'

Reviewer Comment:
Line 6: thermodynamical should read thermodynamic

Author Response:
Thanks.

Manuscript Changes:
Text changed: 'thermodynamical' to 'thermodynamic'.

Reviewer Comment:
Line 8: "little" should be replaced with "only a small"

Author Response:
Thanks.

Manuscript Changes:
Text changed: 'little' to 'only a small'.

Reviewer Comment:
Lines 11-12: here can you draw a connection between large aerosol variation and the mode of ice nucleation that would be dominant?

Author Response:
Yes, variations in aerosol concentrations have an influence on the dominant ice nucleation mode. This has been noted in the text.

Manuscript Changes:
Text added: '…aerosol concentration do affect the dominant ice nucleation mode, however have only…'.

Reviewer Comment:
Line 15: I would say increasingly probable for temperatures lower than -35 C

Author Response:
Thanks.

Manuscript Changes:
Text changed: 'at temperatures around' to 'at temperatures lower than'.

Reviewer Comment:
Line 18: delete "in order"

Author Response:
Done.

Manuscript Changes:
Text deleted: 'in order'.

Reviewer Comment:
Line 21: water vapour deposits directly to ice – you don't specify ice, you just mention deposition of water vapour which doesn't include a phase transition.

Author Response:
Thanks.

Manuscript Changes:
Text added: '…deposited as ice directly…'

Reviewer Comment:
Line 25: "have long" should read "has long"

Author Response:
This refers to both traditional contact and inside-out freezing, therefore the plural form is needed here.

Manuscript Changes:
None.

**Page 2**
Reviewer Comment:
Line 4: if talking about recent reviews then *Kanji et al.* [2017]is the most recent one that discusses the same topics mentioned in this paragraph.

Author Response:
The reference was included in this paragraph.

Manuscript Changes:
Text added: 'Kanji et al. (2017) present a detailed overview of the latest ice nucleation research.'

Reviewer Comment:
Line 13-20 can be one paragraph. No need to have two paragraphs

Author Response:
No problem.

Manuscript Changes:
Paragraphs combined.

Reviewer Comment:
Line 33 " delete "to this"

Author Response:
Ok.

Manuscript Changes:
Text deleted: 'to this'

Reviewer Comment:
Line 34, doesn't the study by [*Spichtinger and Cziczo*, 2010] deserve mention here, since they looked into the competition between homogeneous freezing and heterogeneous ice nucleation (deposition nucleation).

Author Response:
This reference fits better in the paragraph discussing model results. It's been added there.

Manuscript Changes:
Text added: 'Spichtinger and Cziczo (2010) used a model to show there is competition between heterogeneous and homogeneous ice nucleation, which is influenced by thermodynamic and microphysical conditions.

**Page 3**
Reviewer Comment:
Line 5: thermodynamic conditions

Author Response:
Done.

Manuscript Changes:
Text changed: 'thermodynamical' to 'thermodynamic'.

Reviewer Comment:
Line 10: supersaturation is one word and everywhere else in the paper

Author Response:
Thanks.

Manuscript Changes:
Text changed: 'super-saturation' to 'supersaturation'

Reviewer Comment:
Line 15-18 should be in the introduction – doesn't fit in the model description section

Author Response:
The authors believe this paragraph fits better where it is, as it serves as an introduction to the model setup discussion.

Manuscript Changes:
None.

Reviewer Comment:
Line 30-31: dust size distribution at Jungfraujoch from 0.1 – 100 μm, I find that a little hard to believe. Do you have better reference for that? There are a lot of papers published on aerosol and INP properties at the Jungfraujoch that may give you a representative size distribution. I didn't think 100 um dust particles would make it from North Africa to central Switzerland, or at least not in any significant proportion.

Author Response:
The reviewer is correct, the largest aerosol sizes are not represented in the size distribution shown in Figure 1 of the manuscript. Aerosols in concentrations significant enough to be considered have a maximum size could be up to 30 um. But for mathematical convenience a maximum of 100 um was chosen in order to define the size distribution with log-spaced bins. This is clarified in the manuscript.

Manuscript Changes:
Text added: 'Aerosol concentrations at sizes larger than about 30 um are small enough as to be considered zero. The upper bound in the aerosol size distribution is only for mathematical convenience.'

Reviewer Comment:
Line 32-33: If dust is not removed by precipitation in the model, which should be one of the key removal processes – how does this affect ice nucleation in further time steps, i.e. dust that was at lower altitudes that did not activate (because T is not low enough) but also did not get removed by precipitation could then be available to be lofted or for convective uplift for next time steps? Wouldn't this positively bias the role of dust as INP in the model runs?

Author Response:
Generally it should be expected that by not including aerosol removal processes, the INP concentrations should be over-estimated, however the over estimation is expected to be small. The dust aerosol number concentration is orders of magnitude larger than the number concentration of INPs. This means tha any aerosol depletion will make a very small difference to the total dust number concentrations. Furthermore, in the convective and orographically forced cases, aerosols from below the cloud would be transported into the cloud, providing an external source which would constantly replace any depleted aerosols. In the stratiform case, the maximum INP concentrations are even lower than the other cases, and hence still much lower than maximum aerosol concentrations. Even in the event of no entrainment of new aerosols in to the cloud, un-nucleated aerosols within the cloud are still abundant.

This has been addressed in the revised version of the manuscript.

Manuscript Changes:

Text added: 'The aerosols are not removed by precipitation or sedimentation in the model. This simplification is not expected to have a significant effect on the formation of INPs. The maximum number concentration of aerosols is orders of magnitude larger than the maximum INP concentrations, as shown later in this manuscript. Therefore, any removal of aerosols will make a very small difference to the total number concentration. Furthermore, in the case of convectively or orographically forced clouds, entrainment of new aerosols into the cloud adds a source of aerosols to off--set their removal. As for the stratiform case, a factor of 2 overestimate due to not depleting aerosols was found in a previous modelling study (Paukert and Hoose, 2014).'

**Page 4**

Reviewer Comment:

Line 8-11. The authors state that contact freezing of rain drops is not considered but in the every same sentence say that rain drops collect many particles through collision/coalescence processes. But why should this be a reason for contact freezing not to be considered. Perhaps the collision processes could lead to freezing rather than coalescence? Also, in a recent study it was shown that a deliquesced surface of an aerosol particle colliding with a droplet can also induce contact freezing [*Niehaus and Cantrell*, 2015], therefore potentially enhancing the contribution of contact freezing

Author Response:

Paukert et al. (2017) show that freezing of rain drops can be important in the immersion mode in the core of convective clouds. Therefore contact freezing of rain drops may not be efficient, however this has not yet been explicitly shown. For colder or thinner clouds without significant collision-coalescence, freezing of rain drops may not be important. The authors note that no simple parameterizations for this process exist yet, implying it's implementation in these simulations would be a large computational burden. This has been clarified in the revised version of the manuscript.

Manuscript Changes:

Text added: 'Since rain drops collect many particles through collision-coalescence they may be important for freezing in the immersion mode, depending on cloud type (Paukert et al, 2017). However simple parameterizations for this process do not exist, limiting applicability of rain freezing through the immersion mode. Furthermore, Niehaus et al. (2015) show that these deliquesced aerosol particles can initiate additional contact freezing.'

Reviewer Comment:

Line 14-15: this is 50% by number I assume? Please specify

Author Response:

Correct. This has been clarified in the manuscript.

Manuscript Changes:

Text added: '…50\% of the total number of dust aerosol…'

Reviewer Comment:
Line 17: Depletion of immersed aerosols is not taken in to account in these simulations? Does this mean that aerosols that get immersed in cloud droplets are still available for CCN activation in further time steps? How does this impact the results obtained? Given that 50% of your aerosol (by number I assume) is immersed, how would this influence the outcome of the simulations if they are assumed to be available for CCN activation/immersion freezing in subsequent time steps?

Author Response:
The CCN scheme employed here uses the supersaturation to define CCN concentrations representative of continental conditions, as stated on page 3, lines 9 – 11. Since CCN activations is independent of aerosol properties, it will not be influenced by not depleting aerosols. However the INP concentrations, and resulting ice crystal number concentrations will be influenced. This has been clarified in the manuscript.

Manuscript Changes:
Text added: '…not taken into account in these simulations, which has been shown to cause an overestimate of the ice crystal concentrations by a factor of 2 for an arctic stratocumulus cloud (Paukert and Hoose, 2014).'

Reviewer Comment:
Line 19: Replace "so" with "therefore" so is colloquial

Author Response:
Done.

Manuscript Changes:
Text changed: 'so' to 'therefore'.

Reviewer Comment:
Line 30: Has CAPE been defined before?

Author Response:
The definition of CAPE was added.

Manuscript Changes:
Text added: '… a convective available potential energy (CAPE) of …'

Reviewer Comment:
Line 9: x-hour simulation? Perhaps I misunderstood something?

Author Response:
Sorry, that was just a typo.

Manuscript Changes:
Text changed: 'x' to '9'.

**Page 5**
Reviewer Comment:
Line 28-30. I am confused by the wording and reasoning in this sentence. The smaller size and droplet number inhibit INP formation? Perhaps the authors meant "reduce the effectiveness of contact freezing" because of fewer collisions? More INPs would be expected to be active at colder temperatures, but it is fathomable that if collisions do not occur in the first place, then the role of contact freezing would be limited. Perhaps clarify, but the wording "formation of INP" sounds incorrect to me.

Author Response:
The reviewer is correct in suggesting the wording could be better. Contact freezing is more effective at larger sizes due to the collection kernel, which depends on $(r + a)^2$. This has been clarified in the manuscript.

Manuscript Changes:
Text added: '…reducing the effectiveness of contact freezing since the contact freezing collection kernel strongly favours large aerosol--large droplet interactions.'

**Page 6**
Reviewer Comment:
Line 1-3: is there a chance to discuss here or comment on whether homogeneous freezing was suppressed or its initiation was suppressed because of the formation of ice heterogeneously and potentially depleting water vapour. This is hard to deduce from the way the figures are presented. One could discuss size of ice crystals here as well. Perhaps this is lower in the manuscript under temporal distribution..

Author Response:
Some studies do suggest that, in the presence of large INP concentrations, homogeneous freezing could be inhibited. Our simulations show that in the cirrus regime deposition nucleation contributes very little to ice formation, despite the high number concentration of aerosols in this region. The difference between the concentration of homogeneously formed ice and deposition nucleation INP is several orders of magnitude. This strongly indicates that deposition nucleation is not suppressing homogeneous freezing.

Manuscript Changes:
Text added: 'Some studies do suggest that, in the presence of large aerosol concentrations, homogeneous freezing could be inhibited by heterogeneous INP formation (Phillips et al., 2008). The results presented here show that in the cirrus regime deposition nucleation contributes very little to ice formation, despite the high number concentration of aerosols in this region. The difference between the concentration of homogeneously formed ice and deposition nucleation INP is several orders of magnitude. This indicates that deposition nucleation is not suppressing homogeneous freezing.'

Reviewer Comment:
Line 5-10: So you have droplets available for collisions with INPs at RH below 80% how long do they survive and how concentrated are they if they are surviving as droplets at such low RH?

Author Response:
The lifetime of the droplets can be calculated using Equation (3.14) from Houze (1993), ignoring curvature effects and impurities in the droplet. For a 10 um droplet with an environmental RH of 80 %, complete evaporation would happen in 5.7 seconds. For the same droplet exposed to RH of 60 %, evaporation would occur in 2.8 seconds.

Hande et al. (2017) show that in a simulation of a mixed phase deep convective cloud, droplets can be present at temperatures warmer than about 260 K with concentrations anywhere between $10^4$ and $10^8$ m$^{-3}$. In this region, the RH is mostly greater than about 60 %.

These points have been clarified in the manuscript.

Manuscript Changes:
Text added: 'The lifetime of droplets can be calculated using Equation (3.14) from Houze (2014), ignoring curvature effects and assuming pure spherical droplets. A 10 um droplet exposed to relative humidity of 80 % at 260 K should completely evaporate in 5.7 seconds, decreasing to 2.8 seconds at relative humidity of 60 %. Furthermore, Hande et al. (2017) show that in a deep convective cloud, droplets warmer than about 260 K can have number concentrations up to $10^8$ m$^{-3}$. These two points indicate there should be high numbers of droplets available for collisions within a few seconds before evaporating.'

Reviewer Comment:
Line 17-20: I don't get the reasoning here. The RH being high in the mid troposphere should warrant deposition and homogeneous freezing taking place? I would imagine high is a relative term here. 60-70% is high compared to what? So 60-70% RHw is low for homogeneous freezing to take place, so not the result being referred to here is not surprising. Based on lab studies, I am not surprised that deposition nucleation is not contributing at such low RHs either. Generally RHw 70% at about 223 K is usually required for deposition nucleation (or pore condensation and freezing).

Author Response:
The 60-70% figure was calculated from the initial profile of the stratiform cloud shown in Figure 2, and it is high relative to the initial profiles used in the other cases. Generally speaking, it is also high for other thermodynamic profiles used to test the stratiform case (not shown). Despite these higher levels of moisture in the mid-troposphere, compared to other profiles, there still is no deposition nucleation of homogeneous freezing taking place. This was the point the authors were trying to get across, and it's been clarified in the revised version of the manuscript.

Manuscript Changes:
Text added: 'Although the relative humidity in the mid--troposphere is high (around 60-70%) compared to the other profiles shown in Figure 2, homogeneous freezing and

deposition nucleation do not contribute to ice formation.'

**Page 7**

Reviewer Comment:

Line 16-17: it is not clear to me how the authors come to this conclusion about the precipitation. Is this in reference to the total precipitation over the course of the simulation for both cases being compared, or comparing precipitation at a given time stamp? Or the precipitation at the end of the simulation?

Author Response:

This is simply from the bottom panel of Figure 8. The dashed (dotted) lines for the total precipitation are for higher (lower) aerosol number concentrations, and the precipitation throughout the simulation is consistently lower than the normal aerosol conditions. This has been clarified in the revised manuscript.

Manuscript Changes:

Text changed: 'precipitation' to 'domain mean acumulated precipitation throughout the simulation'.

**Page 8**

Reviewer Comment:

Line 1-10: I like the table with the relative contributions of each freezing mode. I assume this is the total contribution over the course of the simulation. However, it would be nice to see more interesting versions, for example you could consider just quantifying the relative contributions up to the point of precipitation initiation – if you just consider the simulation until precipitation starts, could you say something about how much each mode contributes to initiation of precipitation?

Author Response:

Quantifying the contribution before the onset of precipitation is a nice idea, and this has been added to the manuscript. Thanks.

Manuscript Changes:

Table 1: Rows for Homogeneous, Deposition, Immersion and Contact freezing contributions before precipitation onset have been added.

Text added: 'Furthermore, the contribution of each mode until the onset of precipitation ($>$ 0.05 $kg\,m^{-2}$) is shown.'

Text added: 'Leading up to the onset of precipitation, contact plays a dominant role in the semi--idealised convective case and the orographic cloud case. This is since contact nucleation is often the first ice formation mechanism activated, and in these two simulations contributes significantly at early stages of cloud formation. In the other two cases, immersion freezing contributes only slightly more than the simulations with normal aerosol concentrations.'

**Page 9**

Reviewer Comment:
Line 15: Reference format?

Author Response:
Thanks.

Manuscript Changes:
Text changed: 'Paukert et al. (2017)' changed to '(Paukert et al, 2017)'.

Reviewer Comment:
Line 15-18: I assume you are talking about different aerosol species. In this work you have tested dust parameterizations, but one could easily predict how this would change if an organic aerosol parameterization for ice nucleation or soil dust one was used? This should simply make the ice nucleation more or less effective (i.e. lower Ns in the case of organics compared to dust for a set T for example). Couldn't a quick statement from a simulation be made about that in this paper. At least the contribution from soil dust would have been interesting to see here.

Author Response:
This wouldn't be a simple linear scaling, rather the temperature dependence would change, with unknown implications.  A short description of the possible contribution of biological aerosols, amongst others, has been added to the manuscript to address this point.

Manuscript Changes:
Text added:  'Hoose and Möhler (2012) show that biological aerosols have a warm onset temperature in the immersion mode, and given that certain biological aerosols can have large INAS densities at these warm temperatures (Murray et al., 2012), this could represent an important contributor to ice nucleation.  A similar distinction between different dust species could also be made, since soil dust, for example, is more ice active in the immersion mode (Steinke et al., 2016).'

Reviewer Comment:
Line 23: homogeneous freezing accounts for 6% of INP concentrations? Do you mean ice crystals, I didn't think homogeneous freezing was associated with INPs?

Author Response:
The reviewer is correct, homogeneous nucleation isn't associated with INPs.

Manuscript Changes:
Text changed:  'INP' to 'ice crystal'.

Reviewer Comment:
Line 26: steady state being referred to in terms of which quantities?

Author Response:
This statement refers to the production of INPs. This has been clarified in the revised manuscript.

Manuscript Changes:
Text added: 'INP formation in the…'

**Page 10**
Reviewer Comment:
Line 3: "on" the dominant ice nucleation mode?

Author Response:
Thanks for picking up on that.

Manuscript Changes:
Text changed: 'of' to ''on.

**Figures:**
Reviewer Comment:
Are Figures 1-6 for high or low dust concentrations?

Author Response:
These figures are for the normal dust concentrations, ie the solid line in Figure 1. This has been clarified in the figure captions.

Manuscript Changes:
Figure 3 – 11 captions text added: '…for normal dust concentrations'.

Reviewer Comment:
Figure 1: nm or µm is more intuitive. I do acknowledge you want to stick to SI units.

Author Response:
No problem.

Manuscript Changes:
Figure 1: Axis units changed from m to um.

Reviewer Comment:
Figure 3. Homogeneous freezing sets in at about 8km but you have quite a number of liquid droplets above this altitude, are these meant to be conc. solution droplets because your RHw isn't that high? Can you please clarify the existence of liquid droples here? Referring to the top 2 panels in the right column.

Author Response:
This is actually an artifact in the simulation. Droplets greater than 10 um at such a cold temperature would freeze very quickly. Therefore, these anomalous droplets have been masked from the data, and the diagrams re-plotted.

Manuscript Changes:
Figures 3 and 4: Artifacts in cloud liquid number and size plots masked.

Reviewer Comment:
Figure 4: Similar to Figure 3: should there be any droplets when homogeneous freezing has kicked in? Also for the mean cloud drop radius a small comment, more ticks on the scale could be helpful.

Author Response:
This point has been clarified in the above changes to the manuscript.

Manuscript Changes:
None.

Reviewer Comment:
Figure 5: In all these panels, is it possible to zoom into the orographic cloud more and reduce white space .. i.e. there is nothing to show for the altitudes above 3 km – it would give a more clear picture of the orographic cloud. The parameterization limit lines are shown in the Figures 3 and 4, so the reader can be referred to the same limits in Figure 5.

Author Response:
The authors wanted show the white space to explicitly show that there is no deposition nucleation or homogeneous freezing occurring at higher altitudes. Zooming into the region below 3 km would not provide any additional benefit since the cloud is very homogeneous in this region.

Manuscript Changes:
None.

Reviewer Comment:
Figure 6: Same comment as Figure 5.

Author Response:
The authors have already addressed this comment..

Manuscript Changes:
None.

Reviewer Comment:
Figure 10. Is this figure referred to in the text? I don't think so. Also, why the differences in total water content for the high dust case and lower precip for the lower dust case? Perhaps explain this a bit more in the text.

Author Response:
The figure is referenced in the two paragraphs before the discussion section, and has been modified to remove the total water plot.

Regarding the difference in total water content and precipitation, the largest difference is lower cloud amounts above 2 km in the high aerosol environment. This results in lower total water content, however the droplet size and number concentrations are not strongly affected by changes in the aerosols concentrations.

The simulation with lower dust conditions results in very small changes to the simulated cloud properties, therefore there is no strong change in precipitation amount.

Manuscript Changes:
Text added: 'The simulation with higher dust concentrations shows about 25 % more precipitation, despite minimal changes in droplet size and number concentration.'

**References:**
Kanji, Z. A., L. A. Ladino, H. Wex, Y. Boose, M. Burkert-Kohn, D. J. Cziczo, and M. Kramer (2017), Overview of Ice Nucleating Particles, *Meteorological Monographs*, *58*, 1.1-1.33, doi:10.1175/AMSMONOGRAPHS-D-16-0006.1.

Niehaus, J., and W. Cantrell (2015), Contact Freezing of Water by Salts, *J. Phys. Chem. Lett.*, *6*(17), 3490-3495, doi:10.1021/acs.jpclett.5b01531.

Spichtinger, P., and D. J. Cziczo (2010), Impact of heterogeneous ice nuclei on homogeneous freezing events in cirrus clouds, *J. Geophys. Res.-Atmos.*, *115*, D14208, doi:10.1029/2009jd012168.

**REVIEWER #2:**

Review of "Partitioning the primary ice formation modes in large eddy simulations of mixed–phase clouds" by Hande and Hoose

The authors have conducted LES for a few typical cloud cases to systematically investigate which ice nucleation modes dominate. The study has some interesting findings, such as, immersion freezing dominates and contact freezing also contributes significantly in all cases. At colder temperatures, deposition nucleation plays little role, and homogeneous freezing is important; the temporal evolution of the cloud determines the dominant freezing mechanism; precipitation is not correlated with any one ice nucleation mode, instead occurs simultaneously when several nucleation modes are active. However, I have some major

concerns which could impact the key conclusions of the paper. Addressing these concerns would make a more solid paper.

Major comments:

Reviewer Comment:
(1) The results could be dependent on how dust aerosol particles are treated in the model (i.e., prognostic or fixed during the simulation). The information about this is lacking. If dust aerosol particle size distribution is fixed as constant, then cloud properties would not be realistic, especially results of the temporal changes would not be reliable. If prognostic, the relative importance of different modes could depend on the calling order of these modes in the code. For example, if contact freezing is called before deposition freezing and it is efficient, then a lot of interstitial aerosols would be consumed at low-levels (warm temperatures), which would decrease the transported aerosols reaching the cold temperature so the less contribution of deposition freezing will be seen. However, if deposition is called before the contact freezing, the results could be totally changed. The authors did say the dust aerosol concentrations are constant in the vertical dimension. Is it just at the initial time or during the simulation. If it is during the simulation, how is it be realistic? Different assumptions of vertical distributions of dust aerosol particles could also impact the relative contributions of different freezing modes. Perhaps some sensitivity tests on this would like to gain some ideas.

Author Response:
The size distribution and number concentration of the dust aerosols and their vertical distribution is fixed throughout the simulations.

The effect of a variable aerosol size distribution is unknown, and to the authors knowledge has not been investigated with a model before. However it is expected to be small, for the same reason as for effects to due to a variable total aerosol number concentration.

The dust aerosol number concentration is orders of magnitude larger than the number concentration of INPs. This means tha any aerosol depletion will make a very small difference to the total dust number concentrations. Furthermore, in the convective and orographically forced cases, aerosols from below the cloud would be transported into the cloud, providing an external source which would constantly replace any depleted aerosols. In the stratiform case, the maximum INP concentrations are even lower than the other cases, and hence still much lower than maximum aerosol concentrations. Even in the event of no entrainment of new aerosols in to the cloud, un-nucleated aerosols within the cloud are still abundant.

The constant vertical distribution is a simplification, however Hande et al. (2015) found that typical dust aerosol number concentrations during summer over Europe only decrease by about 25 % from near the surface to the tropopause.

Manuscript Changes:
Text added: 'The dust aerosol concentrations are constant in the vertical dimension throughout the simulation. Model results suggest that dust aerosols are relatively constant in the vertical dimension, with only a 25 % decrease of dust aerosol number concentration over

Germany during summer between the low levels and the tropopause (Hande et al., 2015).

The aerosols are not removed by precipitation or sedimentation in the model. This simplification is not expected to have a significant effect on the formation of INPs. The maximum number concentration of aerosols is orders of magnitude larger than the maximum INP concentrations, as shown later in this manuscript. Therefore, any removal of aerosols will make a very small difference to the total number concentration. Furthermore, in the case of convectively or orographically forced clouds, entrainment of new aerosols into the cloud adds a source of aerosols to off--set their removal. As for the stratiform case, a factor of 2 overestimate due to not depleting aerosols was found in a previous modeling study (Paukert and Hoose, 2014).'

Reviewer Comment:
(2) The key conclusions would also be affected by different assumption of the ratio between the immersed and interstitial aerosols. The even ratio used in this study needs some justification (any literature showing such a ratio would do). If there is no justification for this, the generalization of the key results of this paper can be questioned.

Author Response:
The ratio of interstitial to immersed aerosol is simply a multiplicative factor. If this ratio is $\varepsilon$, the number of aerosols available for immersion freezing is $\varepsilon \times N\_aer$, and the number of aerosols available for contact freezing is $(1 - \varepsilon) \times N\_aer$. So yes, the choice of $\varepsilon$ influences the concentrations of aerosols, and hence INPs. However, by setting $\varepsilon = 0.5$ in this study, this influence doesn't impact the results.

This has been clarified in the revised version of the manuscript.

Manuscript Changes:
Text added: 'The even separation of immersed and interstitial aerosols will most likely cause an overestimate of contact freezing, in particular in the updraft where the supersaturation is the highest, and immersion freezing could be more dominant. Unprocessed dust has low CCN activity (Kumar et al., 2011), whereas aged dust is more likely to be immersed. The effect of this uncertainty is, however, expected to be small compared to the orders of magnitude difference in INP number concentrations between the different nucleation modes.'

Reviewer Comment:
(3) More analysis is needed, either to support a key conclusion point or to explain the results (see specific comments # 1, 9, 14, 15). Very often, the authors only describe the results but not go further to explain and understand why.

Author Response:
These points have been addresses individually below.

Manuscript Changes:
None.

Specific comments:

Reviewer Comment:
1) P. 1, Line 10-12, how about changes of rain rate PDF? Also, I did not see significant results presented for the point "Precipitation is not correlated with any one ice nucleation mode". Since this is one of the key results, the corresponding correlation plots should be found easily in the result section.

Author Response:
The correlations coefficients for the domain mean integrated INP concentrations and the domain mean total precipitation were calculated, however they were not significant to any suitably high level. Therefore, they have not been included in the results section of the manuscript. This has been explained in more detail in the revised manuscript.

Manuscript Changes:
Text added: 'Correlation coefficients for the domain mean integrated INP concentrations in each mode, and the domain mean total precipitation were calculated, and the correlation coefficients were not significant to any sufficiently high level of confidence.'

Reviewer Comment:
2) P. 2, L14-15, which study? Also, suggest to use past tense consistently for describing what past work did. Currently, the author mixed the past tense with present tense for these descriptions.

Author Response:
This statement refers to the sentence immediately before it, as should be clear from the context. The tenses in the introduction section were modified to be consistently in the past.

Manuscript Changes:
Past tense used in describing previous studies.

Reviewer Comment:
3) P. 2, L22, I am confused here. How can deep convective clouds always have liquid water at cloud top? This might occur in mixed-phase clouds, but not the deep convective clouds.

Author Response:
The abstract from Ansmann et al. (2009 states:
'Because almost all altocumulus layers (99 %) showed a liquid cloud top (region in which ice nucleation begins), we conclude that deposition and condensation ice nucleation are unimportant processes during the initial phase of altocumulus glaciation.'

Manuscript Changes:
Text changed: 'deep convective' to 'altocumulus'.

Reviewer Comment:
4) P. 2, L32, need to clearly state which studies since no specific studies is mentioned yet in this paragraph.

Author Response:
The idea of this paragraph is to be a general summary of the above studies, in order to lead into the aims of this study. This should be clear, since the authors write: 'The above studies seem to suggest this is the case…'. In order to preserve readability, the authors would prefer not to explicitly state all the references again, since they have been explained in detail throughout the introduction section.

Manuscript Changes:
None.

Reviewer Comment:
5) Section 2, Model description: need more information about dust aerosol simulation in the model, for example, is the dust aerosol size distribution prognostic or fixed during the simulation? See my major comment #1 about the importance of the information. Also, about the vertical distribution, needs to clarify the constant concentration is just at the initial time or during the simulation.

Author Response:
This comment has already been addressed in response to major comment #1.

Manuscript Changes:
None.

Reviewer Comment:
6) P. 4, L15-17, I am not clear how the assumption of the ratio allows the relative concentrations of immersion and contact INPs to be compared independent of this assumption. In addition, is there any measurements in literature showing a ratio of immersed to interstitial aerosols in any place? This ratio could affect the relative contribution of different modes a lot. If there is no justification for this, the generalization of the key results of this paper can be questioned.

Author Response:
The reviewer has already raise this point under 'Major Comments (2)'. The soluble fraction of aerosols is simply a multiplicative factor. If the soluble fraction is $\varepsilon$, the number of aerosols available for immersion freezing is $\varepsilon \times N\_aer$, and the number of aerosols available for contact freezing is $(1 - \varepsilon) \times N\_aer$. These factors are only the same when $\varepsilon = 0.5$, therefore differences in INP concentrations are not due to differences in aerosol concentrations. This has been clarified in the manuscript.

Manuscript Changes:
Text added: '…since differences in INP concentrations will not be due to differences in aerosol concentrations available for nucleation in a given mode.'

Reviewer Comment:
7) P.5, first paragraph, please describe that it is an orographic mixed-phase cloud case.

Author Response:
Ok.

Manuscript Changes:
Text added: '… mixed-phase cloud…'

Reviewer Comment:
8) Figure 3, how to explain the two disconnected layers of liquid in this warm-bubble case? The layer between 9.5-13.5 km is not realistic. Temperatures in this layer could be lower than -38 0C, so liquid particles generally can not survive. So, why is there no liquid between 7.5-9.5 km?

Author Response:
This point was raised by the first reviewer, and as such, has already been addressed.

Manuscript Changes:
None.

Reviewer Comment:
9) p.7, L17, why are both lower and higher aerosol concentrations giving less precipitation?

Author Response:
This is only the case for the semi-idealised deep convective case. The heat bubble showed the opposite: higher and lower aerosol concentrations result in more precipitation. This orographic case showed something different again: higher aerosol concentrations results in more precipitation and lower aerosol concentrations results in lower precipitation. And the fourth case showed something different again: only higher aerosol concentrations give more precipitation, and lower aerosol concentrations have no effect. Therefore there is no single answer to this question, since there is no single clear response. That there is no clear and consistent response to aerosol perturbations is one of the results of this manuscript.

Looking at the microphysical process rates for each case could elucidate the effects of aerosol perturbations, however this is outside the scope of this paper. And once again, since there is no clear response there will be no generalisable conclusions to provide.

This has been clarified in the manuscript.

Manuscript Changes:
Text added: 'The response of the precipitation to perturbations in aerosol concentrations is also complex, and each case exhibits a different response. For the heat bubble, increasing and decreasing aerosol concentrations leads to an increase in precipitation. The opposite is true for the semi--idealised deep convective cloud, where both aerosol perturbations result in a decrease in precipitation. The orographic case shows proportional changes in precipitation

in response to changing the aerosol concentrations, and in the stratiform case the higher aerosol concentrations produce more precipitation, with lower concentrations having no impact. This indicates that, although aerosol concentration plays a role in modifying precipitation, it is not the sole contributor.'

Reviewer Comment:
10) P. 7, last paragraph, the sentence "While the stratiform case has moderate amounts of liquid water, the total precipitation is the lowest amongst all the cases, so much so that the precipitating liquid doesn't decrease the total water" is confusing. The first part of the sentence does not mean much since all the cases are different type of clouds and comparing the correlation between liquid water and precipitation among different types of the clouds does not make much sense physically. Second, I am not sure what you really want to say here for the second part of the sentence "so much so that…".

Author Response:
The total water in the stratiform case reaches a maximum of almost 20 kg m$^{-2}$, compared to over 40 kg m$^{-2}$ for the heat bubble case. Therefore the stratiform case has a 'moderate amount of liquid water'. Total precipitation in the heat bubble case reaches up to 2.5 kg m$^{-2}$, whereas the precipitation for the stratiform case reaches a maximum of 0.2 kg m$^{-2}$, which is 'the lowest amongst all the cases'. This obviously means something because the cloud microphysical properties are different if there's moderate amounts of liquid water, but it isn't precipitating significantly. This should be obvious if one compares the droplet properties in Figure 3 and Figure 6.

The second part of the sentence means that despite there being some precipitation in the stratiform case, it doesn't decrease the total water. This is what's meant by stating 'the precipitating liquid doesn't decrease the total water.' In all the other simulation the precipitation does decrease the total water.

The authors would prefer not to rephrase this sentence.

Manuscript Changes:
None.

Reviewer Comment:
11) P.8, Section 4, this is a discussion session. Table 1 clearly shows main results, so I would suggest to present it earlier (i.e., in the result section).

Author Response:
A new section has been created for the discussion of the domain mean INP concentrations.

Manuscript Changes:
Section Added: '5. Domain Mean INPs'

Reviewer Comment:
12) P9, L9, do you mean the stratiform cloud investigated here has no in-situ cirrus?

Author Response:
Yes, the stratiform cloud is referred to at the start of the sentence.

Manuscript Changes:
None.

Reviewer Comment:
13) P9, L22, what is "a fraction of a percent"?

Author Response:
If the reviewer is interested in the exact contribution of deposition nucleation in each case, under all aerosol conditions, this data is presented in Table 1, and discussed throughout the manuscript. In order to preserve readability, the authors will not repeat the exact values here.

Manuscript Changes:
None.

Reviewer Comment:
14) P9, L25-26, what makes "contact freezing dominated at warm temperatures"? Why immersion freezing is less contributed?

Author Response:
This has been explained in the manuscript already, page 5, lines 24 - 30. The conclusion section is not the place for detailed discussion and explanation of results.

Manuscript Changes:
None.

Reviewer Comment:
15) Last paragraph: Need to explain why the perturbation in aerosol concentrations (means increase or decrease of aerosols) produced proportional changes in the relative contribution of immersion freezing INPs and the relative contribution of the other modes decreased the convective cases. It is especially important to understand how all the other modes are decreased for both increasing and decreasing aerosols. Also, what makes the different results of aerosol impacts among the convective, orographic, and stratiform cases?

Author Response:
Immersion INP number concentrations scale directly with the dust aerosol concentrations, whereas contact INP number concentrations are more complicated due to the dependence on droplet properties and relative humidity. There could also be complex feedbacks present, where changes in the amount of ice produce changes in latent heat release. This would have

an impact on the amount of liquid condensate and hence the dominant ice nucleation mechanism.

Manuscript Changes:
Text added: 'There could also be complex feedbacks present, where changes dust aerosol concentrations change the amount of ice produce, which in turn changes the latent heat release. This would have an impact on both the amount of liquid condensate and also the dominant ice nucleation mechanism.'

Minor comments,

Reviewer Comment:
P. 1, L6, "in each case" should be "between the cases".

Author Response:
The alternative phrasing suggested by the review is equivalent. But it will never-the-less be adopted.

Manuscript Changes:
Text changed: 'in each case' to 'between the cases'.

Reviewer Comment:
P9, L3, change "an orographic mixed–phase case" to "orographic mixed–phase clouds".

Author Response:
Ok.

Manuscript Changes:
Text changed: 'an orographic mixed-phase cloud' to 'orographic mixed-phase clouds'.

Reviewer Comment:
16) P9, L6, incomplete sentence: "There is a fundamental difference between cirrus produced in different dynamical environments", between cirrus and what

Author Response:
The sentence is not incomplete. The difference being referred to here is between cirrus produced in one dynamical environment, and cirrus produced in another dynamical environment. Using the plural 'environments' makes this distinction clear.

Manuscript Changes:
None.

**Author Modifications:**

In Figures 7 – 10, the total water contains the vapor phase, and as such can't be used to understand the evolution of cloud liquid water. Therefore plot of total water in the bottom panels have been removed, and references within the text to the total water have been removed.

---

## Referee Report (RR1)

The authors have addressed most of my comments well, but some of the comments need further clarification. See below,

(1) About my previous major comment #2, I do not think the authors directly addressed my comment. The question is why the INP for immersion and contact freezing should be set to the same. Is this the reality? Any justification from observations?

(2) The author did not address the first part of the specific comment #1. I was asking about the results of rain rate PDFs since the authors only looked at the total precipitation.

(3) The author did not address the specific comment #4. Here is the sentence "Since immersion and contact freezing require the presence of liquid water, they are thought to be the dominant ice formation pathway in mixed phase clouds. The above studies seem to suggest this is the case". This sentence is the start of that paragraph, and so many different studies are discussed in the previous paragraphs. Therefore, the appropriate way to make the sentence clear is either putting references for "the above studies" or replacing "the above studies" by specific references. If the references are many and discussed previously, example references should be put here to help readers to connect with the previous discussion.

(4) For my specific comment #8, the authors responded with "This point was raised by the first reviewer, and as such, has already been addressed". I do not think this is the way to address a comment. You basically asked this reviewer to read another reviewers' comments and your detailed responses to another reviewer. Even if so, there are 14 pages of your responses to the first reviewer and you should at least point out the pages and lines so that I can find the right place. I did a search by searching the keywords "liquid", "layer", etc for this comment and did not find relevant comments from the first reviewer. The authors also indicated there is no text change related to this comment. I am almost sure that explanation about why two disconnected liquid layer exist in the warm bubble initiated convective clouds should be added since this is not something normal. The explanation involves in more analysis as well.

(5) The authors did not address the specific comment #9 well. Yes, the responses of precipitation to increasing aerosol concentrations differ with cases, and the point is to understand why. The authors claimed this is outside the scope of this paper. A common comment of both reviewers was that the paper was lacking in-depth analysis. The reviewer #1 has the exact the same comment about this, i.e., "when there are effects of changing aerosol concentration, these are rightfully stated, but I think the authors could go one step further an explain why this would be expected to have influences (+ve or –ve biases) on the precip amount or total water content". Therefore, "outside the scope of this paper" does not really apply here.

(6) About my specific comment #10, I'd like to reiterate that it is a common base that different types of clouds have different dynamics and microphysical processes, and therefore precipitation efficiency is very different. Therefore, the comparison of rain amount or the relationship of rain amount with liquid/total water between different types of clouds makes no points. In addition, the reason for the precipitating liquid doesn't decrease the total water in the stratiform case might not be microphysical, but entrainment of moisture from cloud top or the change of large-scale forcing, etc. If the

authors want to emphasize this, then you need to provide the reasons to explain it. Otherwise I think you can drop it.

---

## Author Response (AR2)

**Reply by the authors (in blue):**
We would like to thank the reviewer for the careful reading and constructive comments.
**Changes to the text are indicated in red.**

**Comments by the reviewer (in black):**
The authors have addressed most of my comments well, but some of the comments need further clarification. See below,

(1) About my previous major comment #2, I do not think the authors directly addressed my comment. The question is why the INP for immersion and contact freezing should be set to the same. Is this the reality? Any justification from observations?

The partitioning between in-droplet and interstitial dust particles is highly uncertain, and a range of different values has been reported in the literature. This strongly depends on the wettability and size of the dust particles, but also on the cloud properties, e.g. the maximum supersaturation reached in the updraft (determining CCN activation) or the droplet number concentration and size distribution, which determines collision scavenging. As an example form the literature supporting our assumption, Li et al (2011) report that 8.3% of the analyzed cloud residues consisted of crustal dust, compared to 9.15% of the interstitial particles. At the same time, the number scavenging ratio was 0.54, i.e. about the same concentrations of interstitial and in-droplet particles were present, and thus also about the same concentrations of interstitial and in-droplet dust. However, for conditions where dust is among the most active CCN, e.g. after long-range transport, the fraction of in-droplet dust particles is expected to be significantly larger (see also the discussion in Paukert and Hoose, 2014).

The following changes were added to the text:
"The segregation of immersed and interstitial aerosols is treated simplistically in this work, where the ratio of these quantities is pre–defined. In these simulations, 50% of the total number of dust aerosol is defined to be interstitial and hence available for contact freezing, and the remaining 50% is defined to be immersed and available for immersion freezing. This is not necessarily a realistic assumption, however it allows the relative concentrations of immersion and contact INPs to be compared independent of this assumption, since differences in INP concentrations will not be due to differences in aerosol concentrations available for nucleation in a given mode. While some observations support a roughly equal split of dust particles into interstitial and immersed aerosol (Li et al., 2011), we expect this assumption to overestimate the fraction of interstitial dust in conditions where aerosol processing during long-range transport or high supersaturations increase the CCN activation of dust particles (Kumar et al., 2011)."

Additional reference: Li et al, Atmospheric Environment 45 (2011) 2488-2495, doi:10.1016/j.atmosenv.2011.02.044

(2) The author did not address the first part of the specific comment #1. I was asking about the results of rain rate PDFs since the authors only looked at the total precipitation.

The figures below show the PDFs of the accumulated rain rate after four hours (heat bubble convective cloud, orographic cloud, stratiform cloud) and after six hours for the semi-idealized convective cloud. For the low dust simulation of the semi-idealized convective cloud case, the required output was unfortunately not saved. The graphs show a non-systematic behavior: for the heat bubble convective cloud, the increase of the mean rain rate in both the low and the high aerosol case is due to an increase of frequently occurring low rain

rates (<10 mm), while the frequency of occurrence of high rain rates decreases in both cases. In contrast, for the semi-idealized convective cloud case, higher dust concentrations lead to less frequent low rain rates and more frequent high rain rates, resulting in a net decrease of domain-average precipitation. In the orographic cloud case, the upper tail of the distribution is systematically shifted to higher rain rates with increasing dust concentration, and this explains also the shift to higher mean rain rates. In the stratiform cloud case, precipitation is rather homogeneous throughout the model domain, which results in narrow rain rate PDFs. In the high dust simulation, the rain rate is shifted to higher values at all gridpoints.

Given the diversity of the response of the rain rate pdfs, and that they are not easily explained, we have decided not to include these figures into the manuscript.

[Figure]

(3) The author did not address the specific comment #4. Here is the sentence "Since immersion and contact freezing require the presence of liquid water, they are thought to be the dominant ice formation pathway in mixed phase clouds. The above studies seem to suggest this is the case". This sentence is the start of that paragraph, and so many different studies are discussed in the previous paragraphs. Therefore, the appropriate way to make the sentence clear is either putting references for "the above studies" or replacing "the above studies" by specific references. If the references are many and discussed previously, example references should be put here to help readers to connect with the previous discussion.

This was intended to be a summarizing conclusion on the studies discussed in the previous six paragraphs. As suggested by the reviewer, we have now included additional references to the four most pertinent sources as examples:

"The above studies (e.g. Phillips et al., 2007; Ansmann et al., 2009; Hoose et al, 2010; De Boer et al., 2011) seem to suggest this is the case, …"

(4) For my specific comment #8, the authors responded with "This point was raised by the first reviewer, and as such, has already been addressed". I do not think this is the way to address a comment. You basically asked this reviewer to read another reviewers' comments and your detailed responses to another reviewer. Even if so, there are 14 pages of your responses to the first reviewer and you should at least point out the pages and lines so that I can find the right place. I did a search by searching the keywords "liquid", "layer", etc for this comment and did not find relevant comments from the first reviewer. The authors also indicated there is no text change related to this comment. I am almost sure that explanation about why two disconnected liquid layer exist in the warm bubble initiated convective clouds should be added since this is not something normal. The explanation involves in more analysis as well.

We apologize for the imprecise answer. The related answer to the other reviewer was on the top of page 13 of the previous replies.

The presence of (very low concentrations of) liquid water between 9.5 and 13 km was an artifact, resulting from spurious activation of CCN at temperatures below -38°C and immediate freezing. This should not happen in reality, as the reviewers correctly pointed out, because the haze droplets would freeze at these temperatures before water saturation was reached. Upon correction of the condition for CCN activation (now only allowed at temperatures above -38°C), the upper liquid layer disappears. We do not think that this correction should be discussed in the final version of the text, which includes the correct plots. The changes are documented in these replies, which are also archived.

(5) The authors did not address the specific comment #9 well. Yes, the responses of precipitation to increasing aerosol concentrations differ with cases, and the point is to understand why. The authors claimed this is outside the scope of this paper. A common comment of both reviewers was that the paper was lacking in-depth analysis. The reviewer #1 has the exact the same comment about this, i.e., "when there are effects of changing aerosol concentration, these are rightfully stated, but I think the authors could go one step further an explain why this would be expected to have influences (+ve or –ve biases) on the precip amount or total water content". Therefore, "outside the scope of this paper" does not really apply here.

We agree with the reviewer that a further analysis of the effects of perturbed INP concentrations on precipitation is interesting, but still think that this is outside the scope of this paper. In order to understand the response of precipitation, feedbacks on cloud dynamics (e.g. via invigoration effects through additional latent heat release), compensating processes in the warm phase, etc. have to be analyzed. The manuscript however strived to focus on primary ice formation, and we think that an in-depth analysis of the reason for the diverse responses of accumulated precipitation would be distracting. In order to further motivate this, we have added additional text in section 2 (model description) and section 7 (conclusions).

"In order to investigate the sensitivity of ice nucleation to the aerosol size distribution, two additional aerosol size distributions are defined in Figure 1, shown as the dashed lines. Here, the total number concentration of both modes was modified by factors of 10 and 0.1, which simulate high and low dust aerosol number concentrations. These sensitivity studies are

analyzed with a focus on the resulting partitioning into the different ice nucleation modes, e.g. the role of homogeneous versus heterogeneous ice nucleation."

"For the convective cases, perturbation in aerosol concentrations produced proportional changes in the relative contribution of immersion freezing INPs. The relative contribution of the other modes decreased for increased dust concentrations. In particular, homogeneous freezing is nearly entirely suppressed. In contrast, for the orographic case, the relative contribution of contact ice nucleation increased under higher aerosol concentrations, and immersion freezing decreased."

(6) About my specific comment #10, I'd like to reiterate that it is a common base that different types of clouds have different dynamics and microphysical processes, and therefore precipitation efficiency is very different. Therefore, the comparison of rain amount or the relationship of rain amount with liquid/total water between different types of clouds makes no points. In addition, the reason for the precipitating liquid doesn't decrease the total water in the stratiform case might not be microphysical, but entrainment of moisture from cloud top or the change of large-scale forcing, etc. If the authors want to emphasize this, then you need to provide the reasons to explain it. Otherwise I think you can drop it.

As suggested by the reviewer, the discussion of effect of precipitation on the total water path (and the corresponding lines in the bottom panels of Fig. 7-10) has been dropped. This was actually already done for the previous version of the manuscript, and it was on oversight that this reply wasn't updated. We apologize for the confusion.

[revised manuscript text omitted]
 + | $2.94\times10^2$ (0.14) | $2.25\times10^0$ (0.00) | $2.02\times10^5$ (98.07) | $3.68\times10^3$ (1.79) |
| Heat Bubble | $2.77\times10^2$ (1.38) | $3.11\times10^{-1}$ (0.01) | $1.76\times10^4$ (88.31) | $2.05\times10^3$ (10.30) |
| Heat Bubble - | $2.80\times10^2$ (8.73) | $4.11\times10^{-2}$ (0.00) | $2.09\times10^3$ (65.07) | $8.43\times10^2$ (26.20) |
| Heat Bubble Precip Onset | $3.41\times10^2$ (2.52) | $2.14\times10^{-1}$ (0.00) | $1.22\times10^4$ (89.70) | $1.05\times10^3$ (7.78) |
| Deep Convective + | $3.09\times10^1$ (0.12) | $1.47\times10^{-1}$ (0.00) | $2.37\times10^4$ (90.56) | $2.43\times10^3$ (9.32) |
| Deep Convective | $2.35\times10^2$ (6.56) | $2.75\times10^{-1}$ (0.01) | $2.11\times10^3$ (58.95) | $1.24\times10^3$ (34.48) |
| Deep Convective - | $2.51\times10^2$ (28.59) | $8.73\times10^{-2}$ (0.01) | $1.95\times10^2$ (22.18) | $4.33\times10^2$ (49.22) |
| Deep Convective Precip Onset | $3.14\times10^{-2}$ (0.00) | $1.63\times10^{-5}$ (0.00) | $3.97\times10^1$ (3.87) | $9.88\times10^2$ (96.13) |

| | | | | |
|---|---|---|---|---|
| Orographic + | 0 | 0 | $1.68\times10^3$ | $2.76\times10^2$ |
| | (0) | (0) | (85.84) | (14.16) |
| Orographic | 0 | 0 | $1.21\times10^3$ | $1.35\times10^2$ |
| | (0) | (0) | (89.91) | (10.09) |
| Orographic - | 0 | 0 | $7.65\times10^2$ | $4.32\times10^1$ |
| | (0) | (0) | (94.66) | (5.34) |
| Orographic Precip Onset | 0 | 0 | $5.77\times10^2$ | $2.08\times10^2$ |
| | (0) | (0) | (73.50) | (26.50) |
| Stratiform + | 0 | 0 | $6.21\times10^2$ | $1.81\times10^{-1}$ |
| | (0) | (0) | (99.97) | (0.03) |
| Stratiform | 0 | 0 | $1.87\times10^2$ | $4.73\times10^0$ |
| | (0) | (0) | (97.54) | (2.46) |
| Stratiform - | 0 | 0 | $1.85\times10^2$ | $1.49\times10^{-1}$ |
| | (0) | (0) | (99.92) | (0.08) |
| Stratiform Precip Onset | 0 | 0 | $1.76\times10^2$ | $8.37\times10^{-2}$ |
| | (0) | (0) | (99.95) | (0.05) |

Table 1: Temporal and spatial mean INP concentrations ($m^{-3}$) for each case. + (-) indicates higher (lower) dust aerosol concentrations, as shown in Figure 1. The relative contribution (%) of each mode to the total INP concentrations is shown in parenthesis.